# Speech prosody enhances the neural processing of syntax
Giulio Degano [1] ✉, Peter W. Donhauser[2], Laura Gwilliams [3], Paola Merlo [4,5] & Narly Golestani [1,6,7]

Human language relies on the correct processing of syntactic information, as it is essential for successful communication between speakers. As an abstract level of language, syntax has often been studied separately from the physical form of the speech signal, thus often masking the interactions that can promote better syntactic processing in the human brain. However, behavioral and neural evidence from adults suggests the idea that prosody and syntax interact, and studies in infants support the notion that prosody assists language learning. Here we analyze a MEG dataset to investigate how acoustic cues, specifically prosody, interact with syntactic representations in the brains of native English speakers. More specifically, to examine whether prosody enhances the cortical encoding of syntactic representations, we decode syntactic phrase boundaries directly from brain activity, and evaluate possible modulations of this decoding by the prosodic boundaries. Our findings demonstrate that the presence of prosodic boundaries improves the neural representation of phrase boundaries, indicating the facilitative role of prosodic cues in processing abstract linguistic features. This work has implications for interactive models of how the brain processes different linguistic features. Future research is needed to establish the neural underpinnings of prosody-syntax interactions in languages with different typological characteristics.

One of the main properties of human languages is their structured, and not purely linear, nature. Syntactic rules define the syntactic structure, which allows to combine the relatively limited, finite number of words contained in the lexicon into a quasi-infinite number of utterances. This compositional property of syntax has an important role in speech comprehension, since it allows to flexibly combine words into coherent syntactic and semantic units, supporting efficient communication. Thus, the correct processing of syntactic information is critical for successful communication across speakers. Besides syntactic rules, different abstract sources of information can change the way these utterances are processed (i.e., semantic content, context, predictions based on statistics, etc.), but research in the last 20 years has shown that syntax can be also tightly linked to a more physical aspect of speech: prosodic information[1,2].

The suprasegmental information carried by prosody has multiple acoustic markers that can be used to orient the processing of sentence structure, and the semantic interpretation of sentences. These acoustic markers can include variation in pitch or in the energy of the speech signal as well as lengthening of syllables or the insertion of long pauses.

For example, the sentence "Johnny the little boy is playing" has two different meanings that give rise to two different syntactic representations and are expressed by two different locations of the acoustic cues. The lengthening of the syllable *boy* and the placement of an additional pause after it can differentiate between two syntactic structures: one where the sentence is directed at someone called "Johnny" (i.e., "Johnny, the little boy is playing"), and one where "Johnny" is the subject of the verb phrase (i.e., "Johnny, the little boy, is playing") and the noun phrase "the little boy", is an apposition. This example shows how prosodic boundaries can dramatically change how words need to be grouped together in order to successfully communicate concepts. This particular feature of prosodic boundaries can be seen as one of the most evident connections between prosody and syntax as it highlights, for example, a close correspondence with grammatical phrase boundaries[1,2]. The alignment between these two linguistic representations might be key for facilitating the mapping of the incoming speech signal onto a syntactic structure, and disambiguating between the two alternative tree representations that correspond to the two different interpretations.

[1]Department of Psychology, Faculty of Psychology and Educational Sciences, University of Geneva, Geneva, Switzerland. [2]Ernst Strüngmann Institute for Neuroscience in Cooperation with Max Planck Society, Frankfurt am Main, Germany. [3]Department of Psychology, Stanford University, Stanford, CA, USA. [4]Department of Linguistics, University of Geneva, Geneva, Switzerland. [5]University Centre for Informatics, University of Geneva, Geneva, Switzerland. [6]Brain and Language Lab, Cognitive Science Hub, University of Vienna, Vienna, Austria. [7]Department of Behavioral and Cognitive Biology, Faculty of Life Sciences, University of Vienna, Vienna, Austria. ✉e-mail: giulio.degano@gmail.com

The link between prosody and syntax has also extensively been explored in the context of developmental studies, where it has been proposed that prosody assists the acquisition of the first language; this phenomenon is known as prosodic bootstrapping[3,4]. This theoretical framework proposes that infants use prosodic information in the speech signal to help them infer syntactic rules and structures, and thus acquire their first language[4]. Indeed, prosody has been shown to help infants learn syntactic constituency[5], interpret syntactic adjacency[6], and segment continuous speech[7]. Importantly, prosodic bootstrapping has also been shown to occur in different languages, such as English, French, and Italian[8] and in bilingual infants[9]. The significant role of prosody in ontogeny is consistent with its posited role as a precursor of language[10], coming from work showing that newborn babies can already distinguish languages coming from the same rhythmic class as their environmental language from those coming from different rhythmic classes[10,11]. In turn, such a developmentally early sensitivity to prosody likely arises from prenatal processing and learning[12,13], thanks to an auditory system that is established and functional in the last 4–6 weeks of gestation[14–16]. Indeed, prenatal exposure to speech and other acoustic signals is more faithful to slower-changing acoustic information, such as that contained in prosody, due to low low-pass filtering of sound by the amniotic fluid[16–19].

Although the results in children show that syntactic learning becomes less reliant on prosody later in life[20,21], there is evidence of a continuous interaction between these two features even in adults. This is likely due to the fact that prosodic signals can be a reliable acoustic and perceptual scaffold for the online processing of syntactic features[22]. Indeed, it has been shown that prosodic cues such as pitch variation and syllable duration can assist syntax learning more than transitional probabilities[23]. Prosody also helps the initial parsing of sentences[24,25], morphosyntactic dependency learning[7], and speech segmentation[7], in adults.

Biological evidence for a prosody-syntax interaction further supports the importance of prosody for more reliable syntactic parsing. Brain imaging research using electroencephalography (EEG) has shown that changes in pause, pitch, loudness, or duration that mark boundaries between phrases, known as prosodic boundaries[7], affect neural markers of syntactic disambiguation. For example, it has been found that manipulation of the placement of prosodic boundary modulates the evoked responses to the stimuli such as to assist syntactic disambiguation[26,27] and processing of garden path sentences[28–30]. Conversely, research has also shown the influence of syntactic predictions on prosodic processing. For instance, it has been shown that a neural marker of prosodic processing—the closure positive shift[31]—can be elicited by biases generated from initial syntactic parsing preferences, even in the absence of explicit acoustic cues[32]. It is thus possible to hypothesize that the regular interaction between and co-occurrence of these two linguistic features has driven the brain into expecting syntactic and prosodic representations to be coherent even if one of the two is not explicitly present, or when they do not co-occur. In line with this, a recent study has demonstrated that the cortical tracking of naturalistic speech is enhanced when overt and covert prosodic boundaries are aligned with the syntactic structure compared to when they are misaligned[33].

Very few fMRI studies have been done on the interaction between prosody and other, more abstract levels of linguistic information. Seminal studies on the processing of linguistic prosody per se involved comparing different types of filtered speech. These studies showed mixed results in terms of hemispheric lateralization, with some having identified right[34,35] but also some left[36] hemispheric involvement of temporal and frontal-opercular regions in relation to the modulation of pitch (i.e., intonation) during speech processing. The lateralization likely depends on which information is being extracted from the signal or on the control condition[37–40]. The processing of higher-level linguistic features such as syntax is known to be more left-lateralized involving regions such as the inferior frontal gyrus (IFG), and middle temporal gyrus[41–43]. The few studies that have focused on the question of prosody–syntax *interactions* using fMRI showed, for example, that in tonal languages (here: Mandarin), there are shared activations in the

left but not right inferior frontal gyrus IFG between intonational (question vs. answer) and phonologically relevant tonal (word category) processing[39]. Another investigation similarly reported a role of the left IFG in processing linguistically relevant intonational cues; this region's activation was modulated when those cues played a crucial role in sentence comprehension[40].

Although research has shown neural evidence for prosody-syntax interactions, more work is needed to fill the knowledge gap regarding how the brain processes these two linguistic features jointly in more *naturalistic* processing contexts. Most of the studies to date, especially those using EEG, have exploited neural signals or activity emerging from the violation or disambiguation of prosodic or syntactic signals. In the context of EEG studies, this has allowed us to assess the neural markers of time-locked analyses[44], but has limited implications regarding the neural processing of syntactic and prosodic information during naturalistic speech perception. Moreover, controlled studies that make use of expectations or violations can be highly affected by error detection or other processes that are not central to syntactic judgments, again making it difficult to draw conclusions regarding the processing of prosody–syntax interactions per se[29].

In contrast to what can be assessed with well-controlled experimental paradigms, human communication is based on a very complex, multi-dimensional signal. The use of tightly controlled experiments and reductionist, task-based paradigms fall short of an ecological characterization of the neural processes underlying naturalistic language processing[45]. The importance of the generalization of neuroscience research has gained attention in recent years, and the last decade has seen increased use of narratives in experimental designs, to simulate natural speech perception. For this, brain activity is measured during passive listening, to closely simulate the way that speech is processed in day-to-day contexts. In such studies, control over the effect(s) of interest can be applied *after* data collection, at the data analysis stage, thus increasing the flexibility and ecological validity of the experiment[46]. A driving force underlying this paradigm change is the increasingly flexible and powerful computational tools that are now becoming available to model human speech. Indeed, the use of machine learning techniques to statistically assess (and control for) different linguistic features has led to a new line of research that has the aim of modeling complex cognitive and neural functions in ecologically valid settings[47]. Importantly, such model-based encoding and decoding approaches allow results to be interpretable in light of proposed underlying cognitive and/or neural mechanisms[48]. Recent studies have already shown promising results in using linearized modeling approaches for pinpointing the neural signatures underlying the processing of semantics[49–51], syntax[52–54], phonetics[55,56], and acoustics[57–59]. These computational models of the different levels of linguistic representation have however often been investigated in isolation, and very few studies have looked at their possible interactions[50].

In this study, we made use of naturalistic stimuli together with brain decoding[47] to fill a knowledge gap in how the cortical representation of syntax can be modulated by prosody to facilitate speech processing. We used machine learning techniques to compute, on the one hand, the syntactic phrase boundaries, and, on the other hand, the prosodic boundaries of a speech corpus (TED talks).

Here, we first tested whether the strength of the prosodic boundaries present in the speech data was modulated as a function of closing phrase boundaries when producing natural speech (four separate TED talks)[55]. Thus, while not explicitly marking intonational units, we employed prosodic boundary strength since it is a metric that's been shown to positively correlate with intonational units. Then, in an existing magnetoencephalography (MEG) dataset[55] of participants listening to the same TED talks[55], we investigated whether the neural encoding mirrors the prosodic-syntactic interaction we identified in the speech signal. Such a finding would provide biological evidence for possible "prosodic boosting" of syntactic encoding in the adult brain, reflecting a neural mechanism that allows prosodic cues to increase the robustness of syntactic representations, and thus of successful speech comprehension.

## Results

### Stimulus spectrogram analysis

We first analyzed the four TED talks spectrograms, to understand if the speakers used different prosodic cues in correspondence with the syntactic closing phrase boundaries of their utterances. Since the prosodic boundary strength is characterized by positive values and by a skewed distribution, we used a gamma mixture to model the strength distribution. The gamma-mixture model was fitted to the prosodic boundary strength distributions across all four TED talks. This analysis revealed two latent gamma distributions: $\gamma_1$ with shape of 1.1 and rate of 2.8 and $\gamma_2$ with shape of 14 and a rate of 10.7. The gamma-mixture analysis resulted in a boundary between $\gamma_1$ and $\gamma_2$ at 1.01 prosodic boundary strength (dimensionless quantity), thus dividing the words of the TEDLium corpora into content items with high and low prosodic boundary strength (left panel, Fig. 1).

This binarization of the dataset allowed us to uncover the distribution of words with versus without closing phrase boundaries within these two latent prosody distributions. We found that in the presence of weak prosodic boundary strength, the percentage of words without a closing phrase boundary (56%) was greater than the one with a closing phrase boundary (44%), whereas when the speakers used a strong prosodic boundary strength this pattern was reversed (28% for words without a closing phrase boundary and 72% for words with a closing phrase boundary) (right panel, Fig. 1). We then tested for differences in the proportion of word tokens with and without closing phrase boundaries across the two categories of prosodic boundary strength. First, we ran a GLM with a binomial link function between the two levels of prosodic boundary and the presence or not of a closing phrase boundary. This resulted in a significant positive relationship across the syntactic representation and the prosodic boundary ($z$ value = 16.52, $P < 0.001$). Second, to further validate these results, we also ran a linear model using the continuous values of the prosodic boundary strength. To do this, we used a generalized linear model from the Gamma family using the inverse linking function (lme4 package in R 4.2.1[60,61]) and ran the model as done previously. This analysis further confirmed that words with a closing phrase boundary have greater prosodic boundary strength compared to words without a closing phrase boundary ($t$ value = 16.36, $P < 0.001$).

Based on these results, we created three different groups of word tokens that could be evaluated in the brain decoding. The first group comprised of words with a weak prosody boundary, here called "neutral", where a closing phrase boundary could equally be present or absent. A second group included words without a closing phrase boundary and a weak prosodic boundary, as well as words with a closing phrase boundary and a strong prosodic boundary, here called 'coherent', since this group included the combination of syntactic and prosodic information that are more likely to co-occur based on the results of the stimulus analysis, combination that was also predicted theoretically. The third group was composed of words with a

closing phrase boundary and a weak prosodic boundary and words without closing phrase boundary and a strong prosodic boundary. This last combination was called "incoherent", since the stimulus results, consistent with theoretical predictions, suggest that the closing of a syntactic boundary is not generally associated with a prosodic boundary. Tokens across the three groups were not always unique; due to resampling, some were used twice, in order to keep the sample size balanced. Based on these three sets, we assess (1) the contribution of prosodic boundary strength in the encoding syntactic representation in the brain and (2) if the interaction between syntax and prosody that was seen in the stimuli was also mirrored within the cortical representation of the participants (see "Neural analysis").

### Neural analysis

Our analysis of brain activity involved the multivariate decoding of syntactic information from brain activity using a combination of the mne-Python software and custom code. The MEG data were divided into smaller trials based on word offsets, and a logistic classifier was trained to map brain activity to the presence or absence of closing phrase boundaries. Two types of classifications were performed: one using a classical multivariate pattern analysis (MVPA) thus concatenating MEG data across time and channels (Fig. 2), and another performing a temporal classification over the epoched data (Fig. 3). Importantly, the decoder was trained only on neutral combinations to evaluate the neural processing of syntactic information within a weak prosodic context, thus avoiding any inherent bias that was present in the stimuli. By generalizing the performance of the model across the neutral, coherent, and incoherent conditions, it was then possible to quantify and assess differences in the strength of syntactic representations in the brain under various prosodic conditions.

### MVPA decoding

First, we tested whether the neural encoding of closing phrase boundaries was modulated by prosodic strength, in line with our stimulus analysis. Specifically, we test the hypothesis that syntactic classification performance is highest in the "coherent" condition, where the presence of closing phrase boundaries is coupled with strong prosodic cues. In the MVPA analysis of the pre-offset time windows (Fig. 2), we found that the classification performance of coherent and neutral, but not of incoherent trials was greater than chance level (Coherent: Mean AUC = 0.52, $P < 0.001$, MAP = 0.71, 95% HPDI: [0.42 0.91]; Incoherent: Mean AUC = 0.5, $P = 0.134$, MAP = 0.14, 95% HPDI: [0.00 0.41]; Neutral: Mean AUC = 0.51, $P < 0.001$, MAP = 0.43, 95% HPDI: [0.17 0.70]). The decoding of the coherent condition was found to be better than that of the neutral and incoherent conditions (Coherent >Incoherent: $P = 0.008$; Coherent >Neutral: $P = 0.014$). Similar results were obtained in the post-offset time window, where all three conditions performed better than chance (Coherent: Mean AUC = 0.53,

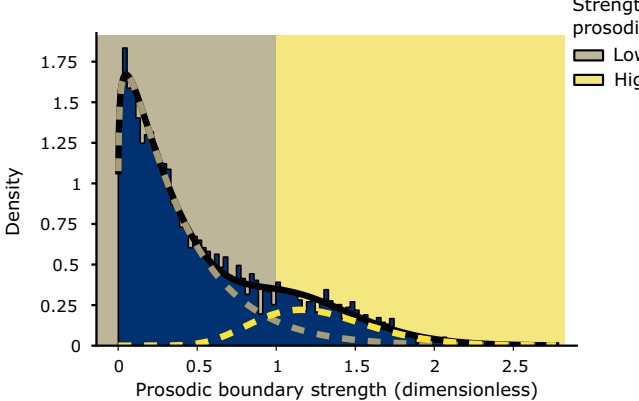

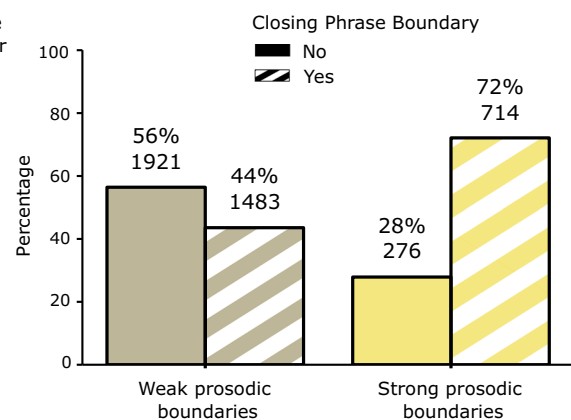

**Fig. 1 | Results of the stimulus analysis.** The left panel shows the two-Gamma Mixture Model that was used to binarize the words according to high or low prosodic boundary strength. The beige line indicates the probability distribution of words with a weak PB, while the yellow line represents the probability distribution of words with a strong prosodic boundary. The right panel shows the frequencies and count of words with and without a closing phrase boundary in the two prosody clusters.

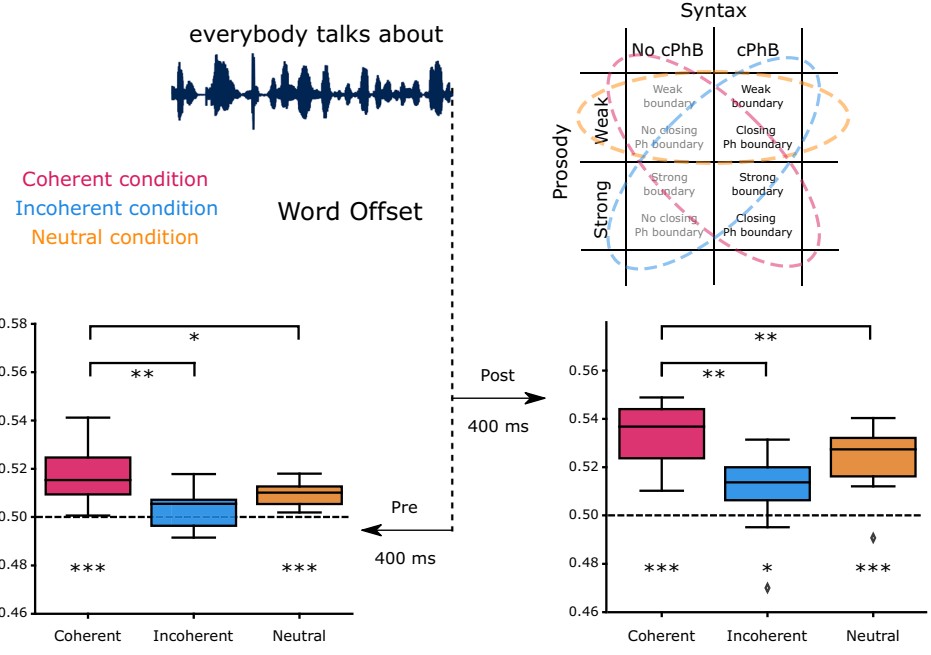

**Fig. 2 | Results of the MVPA.** We decoded the presence of closing phrase boundaries using the MEG activity within 400 ms time windows before and after stimulus offset, for the three generalization sets (coherent in red, incoherent in blue and neutral in orange). The left panel corresponds to the decoding in the pre-stimulus offset time window. In both pre- and post-offset data, the different test sets were also compared to one another; in both cases, the coherent condition was found to be the set with the highest decodability. *$P < 0.05$; **$P < 0.01$; ***$P < 0.001$.

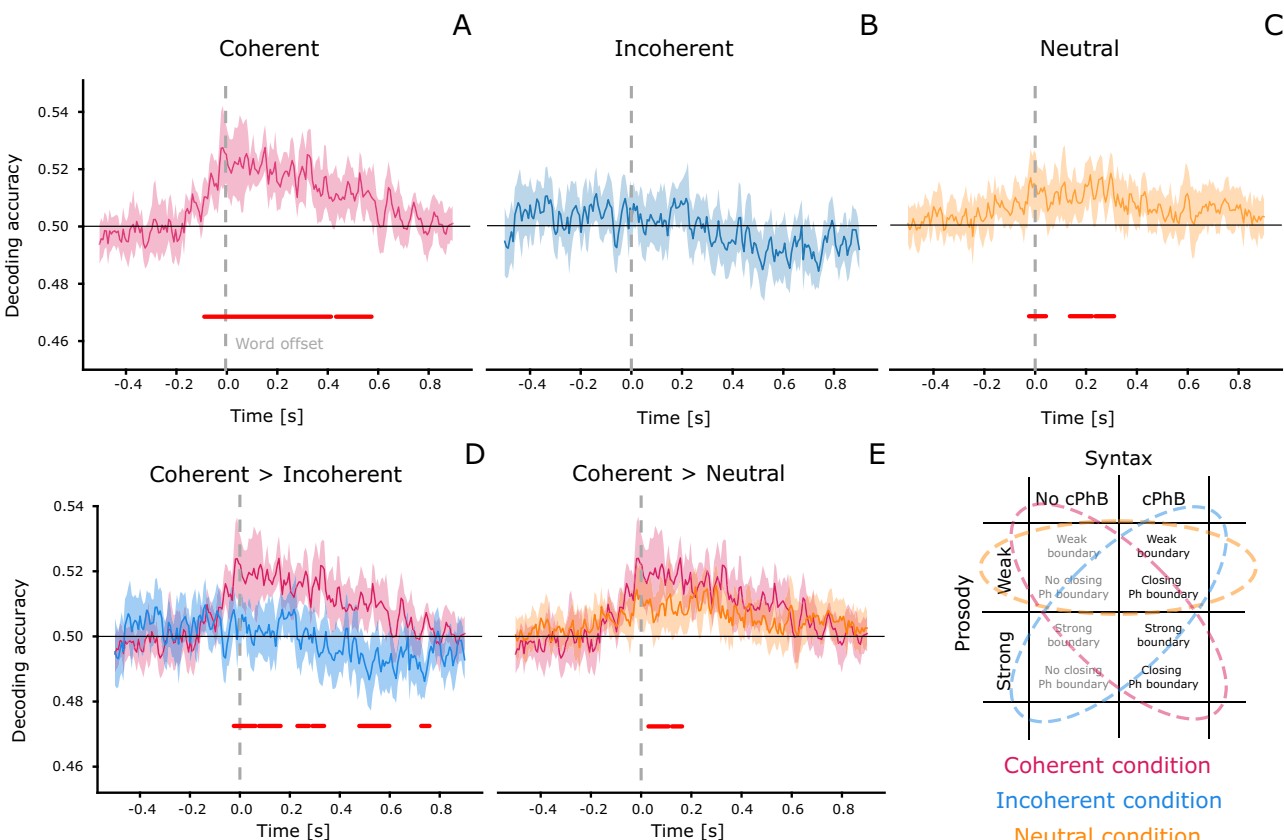

**Fig. 3 | Results of decoding the presence/absence of closing phrase boundary, as a word unfolds.** The upper panel (**A–C**) shows the temporal classification accuracy in the coherent (red—**A**), Incoherent (blue—**B**) and neutral (yellow—**C**) sets. Coherent and neutral sets performed above chance, but both decoding accuracy and the significant time windows differed across these sets. The performance of test sets was then directly compared (**D, E**), to identify the time windows during which the coherent set performed better than the incoherent and neutral ones. We found that in the post-stimulus time window of the first 200 ms, the decoding of closing phrase boundaries in the coherent condition (i.e., containing trials that had a combination of prosodic and syntactic features that was consistent with the pattern that was most frequently detected in the stimuli) was significantly stronger than for closing phrase boundaries with weak or mismatched prosodic boundary strength. Horizontal red lines represent significant results under cluster-based permutation tests.

$P < 0.001$, MAP = 1.00, 95% HPDI: [0.77 1.00]; Incoherent: Mean AUC = 0.51, $P = 0.032$, MAP = 0.62, 95% HPDI: [0.33 0.85]; Neutral: Mean AUC = 0.52, $P = 0.001$, MAP = 0.90, 95% HPDI: [0.64 0.99]) but at different levels, since the decoding of the coherent condition was again better than that of the neutral and incoherent conditions (Coherent >Incoherent: $P = 0.005$; Coherent >Neutral: $P = 0.005$).

## Temporal decoding

To better understand the temporal dynamics of the representation of phrase boundaries, we also employed temporal decoding across the brain activity before and after word offset (Fig. 3). Indeed, this approach offered a more continuous perspective on the temporal evolution of the interactions between the closing of phrase boundaries and prosodic boundary strength. The results of the temporal decoding analysis showed significant encoding of syntactic information in the coherent and neutral test sets (Fig. 3, upper row). The permutation test in the coherent condition revealed two immediately adjacent clusters that together started just before word offsets, and peaked within a 0–200 ms post word-offset time window (Cluster 1: [−0.085, 0.418]s, $P = 0.001$; Cluster 2: [0.438, 0.578]s, $P = 0.013$). In the incoherent condition we did not find any significant effect of decoding, while the neutral condition showed a similar but weaker trend compared to the coherent one, with the three significant clusters mostly spanning the post-offset period (Cluster 1: [−0.024, 0.043]s, $P = 0.021$; Cluster 2: [0.136, 0.223]s, $P = 0.011$; Cluster 3: [0.237, 0.311]s, $P = 0.010$).

In both contrasts comparing test sets, we found better decoding of syntax in the coherent condition, suggesting that higher prosodic boundary strength is associated with boosting of the strength of the syntactic classification (lower row, Fig. 3). The cluster permutation test comparing the coherent with incoherent trials showed significant clusters in the post-offset period (Cluster 1: [−0.024, 0.063]s, $P = 0.012$; Cluster 2: [0.076, 0.163]s, $P = 0.008$; Cluster 3: [0.230, 0.277]s, $P = 0.033$; Cluster 4: [0.290, 0.337]s, $P = 0.031$; Cluster 5: [0.478, 0.599]s, $P = 0.004$; Cluster 6: [0.726, 0.759]s, $P = 0.043$), as did the test comparing the coherent with the neutral condition (Cluster 1: [0.029, 0.110]s, $P = 0.004$; Cluster 2: [0.123, 0.163]s, $P = 0.048$). This demonstrates a consistency between the patterns observed in the neural encoding of the syntax–prosody interaction and in those detected in the speech stimuli themselves.

## Discussion

The purpose of this study was to shed light on the interaction between the representation of prosody and phrase boundaries in the adult brain. We aimed to uncover their relation within utterances produced by different TED talk speakers and within the neural representation of closing phrase boundaries. We tested the hypothesis that the neural processing of sentences at syntactic phrase boundaries is moderated by the prosodic structure of sentences. This would lend support to the idea that prosodic information is used by the brain to facilitate the representation of syntactic relationships during speech processing.

In our stimulus analysis, we showed that speakers modulate their speech output depending on the intended syntactic structure. More specifically, we showed that stronger prosodic boundaries were more likely to be placed on words that had a closing phrase boundary. Conversely, the speech utterances had a relatively weaker if not absent prosodic boundary for words that did not terminate a larger syntactic unit. These results are in line with known linguistic relationships between prosody and syntax in speech, based on evidence that the placement of stress or boundaries in intonational phrases can reflect the syntactic structure of the sentence[1,2]. Our findings support the presence of an interaction between syntactic units and prosodic cues, likely reflecting a mechanism whereby speakers, whether consciously or not, modulate their speech output to help the listener focus on the more salient syntactic relationships between words in order to facilitate comprehension.

Previous such evidence for prosody–syntax relationships has, however, not been investigated in speech corpora using fully automated, large-scale language models for the extraction of prosodic and syntactic features like the

ones adopted here[62,63]. Instead, most of the research exploiting these powerful language parsers has been used for the extraction of prosodic and syntactic regularities from large amounts of data to produce better text-to-speech generators[64,65] or to improve the quality of syntactic parsing[66]. Deep learning approaches can indeed be useful as they indirectly demonstrate a statistical relationship between prosody and syntax, but only a few of these studies have shown links that are linguistically interpretable[67]. Here, we used a data-driven model of language to automatically extract well-characterized linguistic features from a continuous, naturalistic speech signal, and as such our work contributes to a more focused and quantitative analysis of the interface between prosody and syntax than has been demonstrated to date.

In the MEG data analysis, we tested whether the neural processing of syntactic information is modulated by prosody while healthy individuals listened to TED talks. We first trained a decoding model that was free of prosodic biases to learn the mapping of the syntactic information in the neural data. Then, during generalization testing, we contrasted the decoding performance across words belonging to different combinations of prosodic and syntactic information. Two decoding methods were used to assess the relationship. Using MVPA, we tested time windows before and after the offset of each word to see if the neural processing of syntactic information may have differed across different prosody–syntax combinations. The second method made use of temporal decoding, to give us a finer-grained and continuous view of the dynamics of potential phrase boundary and prosodic boundary interactions. Both the MVPA and the temporal decoding analyses revealed that prosodic boundary strength was associated with better classification of the presence or absence of closing phrase boundaries, suggesting that the presence of prosodic cues leads to enhanced syntactic processing. Critically, we found this enhancement to underlie coherent rather than incoherent conditions. We interpret this improved decoding as reflecting that strong prosodic cues serve to enhance the cortical representation of closing phrase boundaries, suggesting a dynamic interaction between prosodic and syntactic sources of information during processing. This in turn supports the importance of bottom-up, acoustic information (e.g., prosody) in shaping higher-level, linguistic representations. On the other hand, the decoder's poor performance on the incoherent condition may be due to conflicting sources of information; in the case where they mismatch, the prosodic cues might either weaken or distort (e.g., introduce noise) the syntactic representation of closing phrase boundaries, which leads to poorer decoding accuracy. To further validate the specificity of this effect, we additionally analyzed the temporal decoding within the four classes separately (see Supplementary Fig. 3 in Supplementary Methods). Results confirmed that the high decoding accuracy of the coherent condition is driven by words having high prosodic boundary strength, accompanied by the presence of closing phrase boundaries. It is worth noting that the decoding approach is based on a static pattern learned in the neutral condition, so increases or decreases in accuracy may indicate either similar or dissimilar processing of syntactic units, respectively. Here, we use the word enhancement as a reflection of a possibly less noisy representation of syntactic boundaries in the coherent condition. Further analyses and studies are needed to better understand the benefits of prosody on syntax processing beyond this effect, for example, by relating differences in neural processing (e.g., decoding accuracy, strength or extent of activation, or other measures) to better behavioral performance. Our finding aligns with previous work showing the relevance of prosody in the modulating brain markers of sentence structures and syntactic operations[26,28–30,32,68], but our work further shows that prosody results in a syntactic representational gain in the neural signal. Moreover, our findings suggest that the temporal co-occurrence or alignment of prosodic cues, grounded in the acoustic signal, and of higher-level, abstract linguistic features can enhance, or boost the neural processing of higher-level linguistic representations. This finding is relevant for evolutionary theories of language, as it could suggest that prosodic cues might have either co-evolved with or even preceded the evolution of syntax, to assist the structuring of human speech. In this context, theorists have already proposed that, as early forms of communication gradually transformed from protolanguages to complex hierarchical structures, prosody likely

served as a foundational element in influencing the evolution of complex grammatical systems[69,70].

Our temporal decoding analysis as well as the MVPA revealed interesting qualitative patterns before and after word offsets. For coherent and neutral conditions, decoding performance was above chance even before word offsets. This suggests the presence of predictive mechanisms, whereby the brain is anticipating the creation of syntactic units before the words are completely heard. Unfortunately, even if the prosodic content is controlled in one of the two conditions that show a predictive effect, we cannot shed light on the nature of this prediction (i.e., acoustic, purely syntactic or a combination of both). In the coherent condition, significant decoding lasts up to 600 ms *after* word offsets, suggesting an additive contribution of acoustic cues (i.e., presence of prosodic boundary strength) to the higher order processing underlying speech comprehension following the representation of a closing phrase boundary[68]. In other words, the presence of prosodic information appears to promote longer-lasting neural processing of syntax, possibly arising from reverberating top-down and bottom-up processing during comprehension.

Our decoding analysis of MEG data limits our ability to localize brain regions involved in the prosody–syntax interface. Previous studies looking at the brain networks modulated by linguistic prosody per se have shown the involvement of regions within the ventral and dorsal pathways spanning the superior temporal lobe and sulcus, premotor cortex, posterior temporal lobe, and IFG, often biased towards the right hemisphere[34,37] but found also in the left, or bilaterally[38,39,71]. Interestingly, two of these areas—the posterior temporal lobe and IFG—have historically been considered to also play a crucial role as hubs for syntactic processing[3,41,42]. It can be thus hypothesized that the posterior temporal lobe and the inferior frontal gyrus (IFG) may be possible loci for the interface between prosody and syntax. Indeed, recent findings seem to indicate the involvement of the left IFG in cases where intonation is used to establish sentence structure[40,72]. Further investigation on the role of linguistic prosody, especially in interaction with syntax, remains to be conducted. This work could, for example, employ functional connectivity analyses, to shed light on how the respective networks interact to support synergistic interaction between different levels of (para-)linguistic processing. This research ought ideally to also integrate more complex models that can disentangle the amount of information conveyed by prosody and its potential to benefit the structuring of human speech. For example, extending our computational approach to data with higher spatial resolution, such as fMRI, could uncover the neural architecture that exploits prosodic information to enhance the cortical representation of syntactic structures.

## Methods
### Participants
We made use of an existing MEG dataset obtained from 11 participants in previously published work (20–34 years; 5 female) 57. All participants signed written consent. The study was approved by the Montreal Neurological Institute's ethics committee (NEU-11-036), in accordance with the Declaration of Helsinki. All ethical regulations relevant to human research participants were followed.

### MEG data acquisition and preprocessing
The MEG signals were recorded using a 275-channel VSM/CTF system, with a sampling rate of 2400 Hz and low-pass filtered at 660 Hz. Eye movements and blinks were recorded using 2 bipolar electro-oculographic (EOG) channels.

Independent component analysis was used to remove artifacts due to eye movement and heartbeats. The MEG signal from each channel was subsequently visually inspected to remove those containing excessive broadband power spectra. For the coregistration between anatomical data and the MEG channels, the head shape of each participant was digitized using Polhemus Isotrak (Polhemus Inc., USA), based on T1-weighted structural MRI data (1.5 T, 240 × 240 mm field of view, 1 mm isotropic, sagittal orientation).

Forward modeling was computed on the MEG data using the overlapping-sphere model[73]. Noise normalized minimum norm operator was used to estimate the inverse solution[74]. Further, singular value decomposition was performed on the resolution matrices defining the source space. For each session and subject, this operation produced a set of M singular vectors reflecting an orthogonal space on which the coregistered data was projected. By taking the 99% cutoff of the singular value spectrum, we defined a low-dimensional space $X_c$ over which the decoding analysis was computed. For further details regarding the data preparation, see ref. 55.

### Speech stimuli
The speech stimuli used during the MEG acquisition consisted of 4 different talks (Daniel Kahneman, James Cameron, Jane McGonigal, and Tom Wujec), extracted from the TEDLium corpus[75]. Seven audio files were created by splitting three TED talks into two parts, while one was kept intact. Audio files lasted 509 s on average, and only one was played during each block of MEG acquisition. There were 7 blocks in total, with an average length of 8.49 min [+/− s.d. of 1.24 min] (see Supplementary Table 1 in Supplementary Methods for more information).

The speech stimuli were forced-aligned to the TED talk text, extracted from the online transcripts using the Montreal forced aligner (MFA[76]. The automatic alignment was implemented with the LibriSpeech corpora[77], available as an MFA model. The output of the alignment was examined and corrected manually using the Praat software[78].

Further stimulus adjustments were done to optimize the following MEG analysis (see Decoding section). First, out-of-dictionary tokens were labeled as <unk> and removed from the set. Second, to avoid positional confounds due to end-of-sentence effects, the last word of each sentence was also removed.

The complete preprocessing of the TED talks produced a stimulus set consisting of 1723 unique words that were subsequently used in the stimulus and MEG data analysis.

### Prosodic and syntactic features in the speech stimuli
Two sets of features were extracted from the text and audio signal to characterize the syntactic and prosodic components, respectively (Fig. 4). The sentences obtained from the online transcriptions of the selected TED talks were tokenized, tagged with their Part-of-Speech label, and finally annotated with an automatic parser. The complete annotation was performed using the pre-trained Spacy pipeline (en_core_web_trf, version 3.2.4, https://spacy.io/Berkeley Neural Parser), with the RoBERTa transformer[63]. After the parsing was computed, we assumed a bottom-up traversal of the tree and counted the number of closing nodes for each word in the dataset. We assume this bottom-up traversal of the tree as previous neurophysiological evidence has highlighted a cortical implementation of a bottom-up parser of the incoming language stream using high-precision intracranial recordings[79]. Based on this traversal, we assigned each word from the stimulus set to two different classes. One class represented the group of words that closed more than one node, and were thus associated with a closing phrase boundary, the other represented the group of words that did not close any phrase or nodes (Fig. 4).

In the second step, we computed a spectro-temporal analysis of the audio files to extract the word duration, the energy, and the $f_0$ information. The time courses of these three features were then combined and analyzed via wavelet transform. The spectrogram obtained after this last step was used to evaluate the maps of minimal power around word offsets to assess the prosodic boundary strength associated with each word. This unsupervised modeling of prosodic information was computed using the Wavelet Prosodic Toolkit to extract the prosodic boundary strength (https://github.com/asuni/wavelet_prosody_toolkit). While we have not manually marked intonational units, we employed prosodic boundary strength as a metric for these units since it has been shown to positively correlate with human annotations[62]. Words that were automatically tagged with prosodic boundary strength equal to 0 were discarded from further analysis as they did not contain meaningful spectral content. The prosodic distribution of

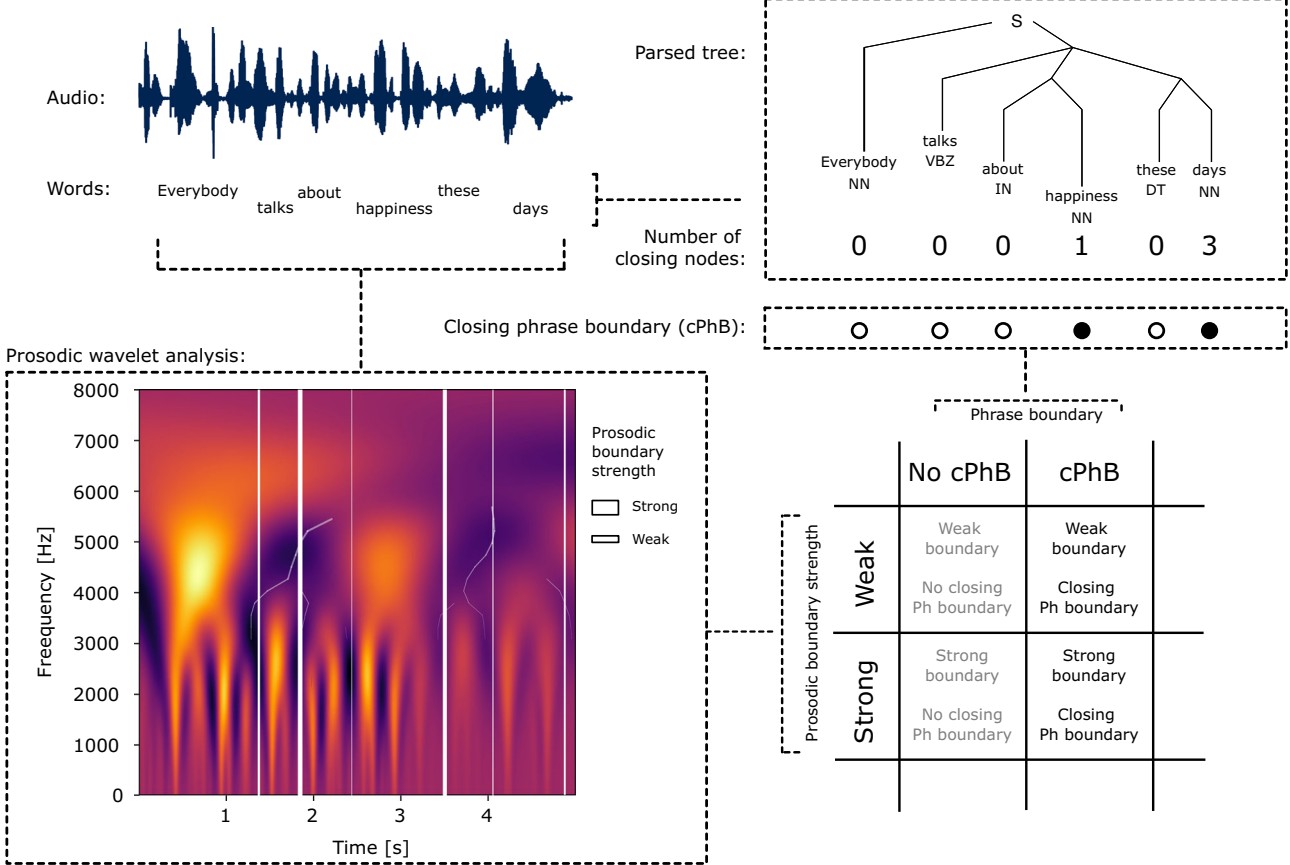

**Fig. 4 | Pipeline for the extraction of the syntactic and prosodic features from the stimuli.** Top right corner shows that we first extracted the constituency trees from the text annotation using Spacy (Capital letters correspond to the POS of the token NN = Noun; VBZ = Verb; IN = Preposition; DT = Determiner). We then marked each word in the text based on the presence (filled dot) or absence (empty dot) of closing phrase boundaries. The bottom left corner shows that we computed a wavelet analysis of the stimulus acoustics to extract the prosodic boundaries (PB—white lines). Prosodic boundaries varied in their strength, depending on the local spectral characteristics of the speech stimuli (thickness of the lines reflects stronger boundaries). Finally, using gamma-mixture models we binarized the prosodic boundaries in two groups, to allow investigation of the prosody–syntax interaction.

the data was subsequently modeled using a two-Gamma Mixture model to cluster the complete stimulus set into two latent distributions. This step allowed us to classify the prosodic content of each word within the binary categories of weak and strong prosodic boundary strength. The final feature matrix Y describing our stimuli was thus comprised of a syntactic feature characterizing the words as having or not having a closing phrase boundary, and of a prosodic feature characterizing the words as having a weak or strong prosodic boundary strength (Fig. 4).

To assess the interaction between prosodic and syntactic information in the stimuli, we first separated all the words present in the dataset based on the weak or strong prosodic boundary strength content. We then compared these two groups regarding their percentage of words that did or did not have a closing phrase boundary. This allowed us to examine whether or not prosodic boundary strength was modulated in relation to syntactic structure during the production of the selected TED talks (Fig. 1).

### Model-based decoding of syntax in the MEG data
The multivariate decoding of syntactic information from brain activity was implemented using a combination of the mne-Python software (version 1.0, https://github.com/mne-tools/mne-python) and custom code. The low-dimensional MEG data $X_c$ was separated into smaller trials by extracting the brain activity before and after each word offset. Trials with an inter-trial offset interval of less than 100 ms were discarded. Following that, a logistic classifier was trained to learn a linear mapping between the brain activity and the presence or absence of closing phrase boundaries. Two different types of classifications were computed. First, MEG data was concatenated across time and channels in the time windows pre-word and post-word offset to assess the influence of prosody, with the prediction that syntactic decoding performance would be modulated by prosodic boundary strength in a way that is coherent with the pattern obtained in the stimuli themselves. Second, a refined decoding analysis of each time point between 400 ms pre-offset and 800 ms post-offset was done, to assess finer temporal modulation patterns of the cortical representation of closing phrase boundaries.

### Training and test datasets
The data to be used for training versus testing was selected based on the data-driven modeling computed on the word stimuli. Following the analysis of the stimuli (see 'Stimulus analysis' subsection of the Results), the training set was created by selecting only words having a weak prosodic boundary strength, including an equal number of words with and without closing phrase boundaries. This allowed us to obtain a classifier that was not influenced by the different levels of prosodic content, and that was syntactically balanced.

The model was assessed using three different test sets, each containing the same number of words from each syntactic category (i.e., with and without closing phrase boundaries) that were not overlapping across sets or repeating from the training sample. The first test set, similar to the training dataset, contained only words that had low prosodic boundary strength (*neutral*). The second set contained words having a closing phrase boundary and strong prosodic boundary strength, and words without a closing phrase boundary and weak prosodic boundary strength (*coherent*). The final set contained words having a closing phrase boundary but weak prosodic

boundary strength, and words without a closing phrase boundary but with strong prosodic boundary strength (*incoherent*). The final training set was composed of 3740 words and the three testing sets were composed of 654 words each. These three different sets aim to characterize syntactic distributions (i.e., presence or absence of closing phrase boundaries) within different combinations of high and low prosodic boundary strength content. Importantly, we expected the three test sets to generate separate brain features or states that could or could not improve the classification accuracies of the classifiers based on the interaction between prosody and syntax. This analysis allowed us to hypothesize stronger syntactic decoding in the test set that was coherent with the results shown in the stimulus analysis (i.e., in the high separability set). Moreover, this procedure allowed us to quantify differences in the strength of syntactic encoding across the conditions, given that they were balanced such that each included the same number of stimuli with versus without phrase boundaries.

### MVPA of pre and post-word offset

We used a logistic classifier (regularization parameter C = 1) to predict the presence of words with or without closing phrase boundaries. This linear mapping was tested by pooling all the brain activity time point x channel in two separate time windows: a 400 ms window before word offset and a 400 ms one after word offset. We hypothesized that the MVPA approach would also show that the encoding of closing phrase boundaries is modulated by the presence of prosodic information, i.e., that we will find evidence for prosodic boosting of syntactic processing.

We evaluated the performance of the model by assessing the area under the curve (AUC) of the classification against chance (50%) on the test sets. The training and testing were repeated by performing tenfold randomized cross-validation.

### Temporal decoding

A single regularized logistic classifier was trained for each time point, and optimized via nested fivefold cross-validation. The regularization parameter $C$ was selected during nested cross-validation via a log-spaced grid search between $10^{-5}$ and $10^{5}$.

The best temporal model obtained after the grid search was then tested by estimating the AUC of the classification (sklearn.metric.roc_auc_score) across each of the different conditions described above: neutral, incoherent and coherent.

The AUC results across these three test sets were used for two different contrasts to assess the effects of prosodic boundaries on the decodability of syntactic phrase boundaries from brain activity. In the first contrast, the coherent condition was tested against the neutral condition, while in the second, we compared coherent against incoherent separability. Critically, while the first contrast allowed us to understand the effect of prosodic content on closing phrase boundaries, the second verified that the prosodic boosting is specific to words with closing phrase boundaries. This strategy allowed us to show that the potential improvement in decoding was not due to a purely acoustic effect, but to an interaction between syntax and prosody.

### Statistics and reproducibility

In the MVPA analysis the differences across the three conditions and against chance level were assessed in a second-level analysis via paired permutation testing with 2048 permutations (permutation_test of mlxtend package). $P$ values of the comparisons were adjusted via Bonferroni–Holm correction. Assumption of a chance-level AUC of 0.5 was verified via a permutation test (see Supplementary Fig. 2 in Supplementary Methods).

To further validate our decoding approach and the replicability of the effects at the individual level, we performed additional prevalence testing analyses following a Bayesian approach[80]. Supplementary Fig. 1 in Supplementary Methods shows the posterior distributions of these tests. Population prevalence Bayesian maximum a posteriori estimate (MAP) values and 95% highest posterior density intervals (HPDIs) were reported in "Results".

Similarly to the MVPA analysis, the temporal decoding results were evaluated using a cluster-based permutation test. The temporal decoding of

the three test sets as well as of the two contrasts were extracted for each participant, and tested against chance (50%) in a second-level analysis. To do this, we ran a nonparametric cluster-level paired $t$ test with 2048 permutations ($2^{\#\,of\,Subjects}$) on the AUC time series obtained from each participant (mne.stats.permutation_cluster_1samp_test).

The permutation tests in the MVPA and in the temporal decoding were computed with the available sample size of 11 participants.

### Reporting summary

Further information on research design is available in the Nature Portfolio Reporting Summary linked to this article.

### Data availability

The decoding data files to reproduce the results using the available code, including to generate the graphs in Figs. 2 and 3, are available at figshare (https://doi.org/10.6084/m9.figshare.23609505). The raw data are available for research purposes only upon request from the corresponding author (giulio.degano@unige.ch) or Peter Donhauser (peter.donhauser@esi-frankfurt.de), because of constraints imposed by the ethics approval under which this study was conducted.

### Code availability

The code for the stimulus analysis and results visualization is available at https://doi.org/10.6084/m9.figshare.23609505.

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

## Acknowledgements

This research was funded by the NCCR Evolving Language, Swiss National Science Foundation Agreement #51NF40_180888. The authors thank Sylvain Baillet for sharing the MEG data, and John Hale and Donald Dunagan for useful exchanges regarding syntax.

## Author contributions

Conceptualization: G.D. and N.G.; methodology: G.D. and L.G.; software: G.D.; formal analysis: G.D., N.G., L.G., and P.M.; investigation: G.D. and N.G.; resources: N.G.; data curation: G.D. and P.D.; writing—original draft: G.D. and N.G.; writing—review and editing: G.D., N.G., L.G., P.D., and P.M.; visualization: G.D.; supervision: N.G.; funding acquisition: N.G.

## Competing interests

The authors declare no competing interests.
