## [Peer review file · Communications Biology]

Reviewers' comments:

Reviewer #1 (Remarks to the Author):

The paper "Speech prosody enhances the neural processing of syntax" by Degano et al. is a paper on the issue how different aspects of speech processing are merged, here merely prosody and left dependency using natural speech as provided by TED talks. The paper reads very well, however, I am not fully satisfied with the grouping of conditions into neutral, coherent and incoherent. Furthermore, I am missing the localization within the brain. Where in the brain are the different aspects of speech processed?

Strong prosodic boundaries were observed twice as often for left dependency words as for no left dependency words (70% vs. 30%). For weak prosodic boundaries there was almost no difference.

While the grouping in 'neutral', 'coherent', 'incoherent' certainly is of a general interest, I was asking myself why a classification according boundary strength and left dependency was not investigated? In the strong boundary condition - which difference can be observed for LD vs. no LD? For LD - what is the effect of the boundary? In case of no LD is the boundary effect the same?

Please check the values - they are identical for weak and high prosodic boundary strength.

"We found that while in the weak prosodic boundary strength cluster, the difference in the proportion of left dependencies was below a 5% frequency shift (p -value of our samples = $7.914e-10 > p$ -value of test distribution = $6.233e-14$), the cluster with high prosodic boundary strength showed a strong bias towards words with left dependencies (p -value of our samples = $7.914e-10 < p$ -value of test distribution = $6.233e-14$)."

For the MVPA decoding - how was the chance level estimated? The theoretical level of 0.5 seems to be not appropriate as the classification sometimes is below 0.5? See Allefeld et al 2016 and Valente et al 2021 on how to deal with measures like accuracy values statistically. Please revise the statistics - and test the conditions before combining them.

Using the minimum norm estimates to compute the brain activity is fine, of course. However, it would be interesting to play around with the different regions separately. In that case, a beamformer solution could be superior as such a solution can be computed for each region separately. The solution is independent of the other regions. One could conduct the MVPA analysis for a set of brain regions known to be involved in processing of prosody and syntax. Would that lead to superior classification results?

minor remarks:

Please use abbreviations for the units in correspondance to the SI, e.g. s instead of sec.

pVal = is usually written as p =

Please present results with the precision motivated by the SNR or some error discussion, e.g. a p-value of 7.914e-10 larger than 6.233e-14 could be more effectively reported as 8e-10 and 6e-14

or the percentage of words without left dependency 55% versus with left dependency 45% ... 31% for words without LD and 69% with LD .. Furthermore, the parameters of the Gamma Distribution - are the five digits really relevant?

Inchoerent -> incoherent

Reviewer #2 (Remarks to the Author):

The study presents a very interesting relationship between prosodic characteristics of spoken language and MEG measures of brain activation while people listen to the same language. The methodology seems appropriate, but I do not have the expertise to judge it. The study used data driven, fully automated classification methods, which I do not know enough about to critique.

First, the results showed a systematic difference in the spectrogram of the speaking data, that shows, using this methodology, a relationship between speakers' production of strong prosodic boundaries and syntactic left dependencies. We know this, but these results, from a novel methodology converge with previous findings. The main results of this study are the relationships between prosodic patterns in speech and MEG patterns in the brains of listeners to that speech. All the combinations of syntax and prosodic patterns, even the ones labeled as incoherent, were significantly related to MEG patterns, however, consistent with the main hypothesis, the relationship was stronger between the brain patterns and the coherent combinations than with the neutral and incoherent. This was true also for the measures of temporal decoding.

So basically, this study has shown, with a different methodology, that prosodic patterns are related to syntactic structure and facilitate brain responses. These results are not novel, but present important converging evidence for the importance of prosody, and acoustic characteristic of speech, for syntactic processing. Given the importance of converging data for psycholinguistics, I believe this paper is worthy of publication, with the caveat that I cannot judge this methodology.

Reviewer #3 (Remarks to the Author):

SUMMARY

This is a fine n = 11 MEG study on the interaction of prosody and syntax in the human brain. The authors report that brain activity during the ending of a syntactic dependency can be decoded better when this ending coincides with a prosodic boundary.

The findings have potential for publication. I have some comments on the conceptualization, framing, and motivation that might further improve the value of the current work.

MAJOR

(1) Assumptions versus facts in the motivation of the current study: The authors claim that the hierarchy of syntax is a fact, which is not the case. Rather, hierarchy is assumed by certain theories in (psycho)linguistics, but not others. This should be acknowledged and the text, in part, should be rephrased / toned down. Some example comments on this:

1/16: "hierarchically" This is not a fact, but a theoretical assumption. Please rephrase ("many linguists assume that" or "one prominent theory of syntax assumes"). I also do not understand how hierarchy is required to make language "efficient and robust". What exactly do the authors mean here by "efficient and robust"?

1/32: See my above comment on assumptions versus facts. It is a frequent habit to call hierarchical grammar a fact about language. This is harmful to the neurobiology of language. There are, for instance, many frameworks in (computational) psycholinguistics that do neither assume hierarchy nor explicitly represent it (e.g., cue-based retrieval, dependency locality theory). The authors should rephrase this sentence and stress that the content is based on their belief / theoretical reasoning, not on facts; alternatively, if they insist that hierarchy is a fact, they should cite and make explicit a piece of evidence that shows that hierarchy is a fact, in a way understandable / transparent to readers from (neuro)biology.

1/32ff: Can the authors give one example where the formation of hierarchical dependencies is required to understand someone? They write here that "syntactic rules" are "fundamental for successful communication". An example would help non-expert readers to understand this notion here.

6/156: Not all dependency grammars "create hierarchical syntactic representations". Consider, for instance, Nivre's recent work and the experimental work by McElree on the insensitivity of memory retrieval to order. Please rephrase.

15/355: Please explain to the reader how the current findings speak to the processing of "hierarchical structures". Make this explicit, as currently, I think there is no clear link in the text between hierarchies and prosody and boundaries and dependencies.

(2) Dependency versus boundary: The authors are using the ending of *any* dependency to define the closure of a syntactic unit. I have doubt about this: (1) prosodic boundaries are strong at the end of large syntactic units (e.g., clauses, sentences), but markings at small units are much weaker. (2) Babies bootstrap into syntax from large to small, where only large units can be inferred from acoustics, but small units require statistics. (3) While any prosodic boundary marks a syntactic boundary, not all syntactic boundaries receive a prosodic boundary. (4) Importantly: Not any syntactic dependency coincides with the end of a large syntactic unit. I think that the manuscript in its current form mixes (a) the closure of large syntactic units and (b) the closure of any syntactic dependency. In order to fix this, the authors might think about their definition of syntactic boundary again. I think that there might be a way of defining syntactic boundaries in a more narrow manner that is easier linked to prosody as well, find below some hints on this:

1/23: I am not sure I understand the rationale here. Why would the processing of "prosodic boundaries" affect dependency processing? IPBs mark the on- and offsets of larger syntactic units (clauses, sentences, possibly constituents), but they do not mark dependencies (unless the authors assume that the termination of a dependency is always followed by an IPB, which is certainly not correct in its generality / restricted to head-final constructions and elements occurring at the end of said large syntactic units). Prosodic markings of dependencies are not called prosodic boundaries, these are, for instance, pitch accents that accompany topicalized elements in dependencies. Please make more explicit why and how you think prosodic boundaries relate to dependencies. Edit: From reading the introduction further, I now understand that the authors' work is on the alignment between syntactic and prosodic boundaries (i.e., *phrase*/etc. closure), not dependency processing. Possibly they can rewrite the abstract and first paragraph. Delineating larger syntactic units (such as phrases, clauses, sentences) is very much different from hierarchical syntax (e.g., Christiansen's work, Fodor's work, Frazier's work). There are segments, and then there is fine structure. The prosody–syntax link is likely constrained to the boundaries of large syntactic constituents. The manuscript

should be framed in this way I think, rather than jumping the gun with hierarchy (which does not seem to be the objective here).

2/54: "close correspondence" and "redundancy" This correspondence is not direct and not as strong as the authors want to make the reader believe here (e.g., Truckenbrodt, 1999; Grosejan & Lane, 1979). In particular, prosodic fine structure is debated. It is clear that clauses / sentences / heavy NPs have strong boundaries; beyond these, the evidence is thin. Please be more specific here. There is certainly no full "redundancy".

6/154: "dependency structure" Again, I am not sure how this is supposed to link to prosody. The link between large syntactic units and prosody is where things interact, not in the entire "dependency structure". Phrasing, chunking, clauses, sentences, constituents... How would the authors define the syntactic boundaries of these abstract units based on the "dependency structure"? Maybe the authors could make a figure of an attachment ambiguity to explain to the reader (a) why hierarchy matters, (b) how dependency and prosody come together?

7/177: From the last paragraph of the introduction, it seems to me that the authors assume that "prosodic boundaries" mark all "dependencies" / the termination of all "dependencies". This is quite unusual, in my view. Prosodic boundaries delineate large syntactic units, such as clauses, sentences, heavy NPs. Prosodic boundaries do not mark any individual dependency (see above comments).

7/183: What do the authors mean by "robustness"? What particular aspect of the "representations" is increased in "robustness"? From what has been written, it seems that the authors propose that dependencies are somehow processed more readily when the left-hand element coincides with a prosodic boundary? Again, I am quite confused by what the authors think the function of prosodic boundaries is. To my understanding of the literature, prosodic boundaries mark mostly large syntactic boundaries. So, in my view, the authors should be modeling syntactic boundaries (e.g., of clauses, phrases, large NPs, chunks...) rather than dependencies.

8/198: "This binarization" Again, I am puzzled: Where is the idea coming from that prosodic boundaries mark "left dependencies"? Prosodic boundaries mark *boundaries* of (large) syntactic units. There is literature on prosody marking dependencies (e.g., topicalization, information-structurally driven), but that type of prosody is not prosodic boundaries, but pitch accent / focus etc. The authors' analysis here likely picks up stress / focus rather than prosodic boundaries.

8/218: What is the basis of the idea that "prosodic boundaries" coincide with "left syntactic dependencies"? Prosodic boundaries mark on- and offsets of units, not individual dependencies.

13/294: Framing of study is unclear to me. The authors need to spell out earlier what they think the relationship is between prosodic boundaries and syntactic dependencies. In the current version, I find this link not well motivated. In my view, the broader literature shows that prosodic boundaries mark the boundaries of (larger) syntactic units. Dependencies are not marked by prosodic boundaries. I suspect that what the authors observe in their results is a mixture of prosodic markings of syntactic phrase boundaries (= prosodic boundaries) and prosodic markings of NADs (= pitch accents, intonational focusing). This could be one way of reanalyzing the data: (1) dissociating prosodic boundaries and pitch accents, (2) dissociating syntactic boundaries and NAD onsets. In the current state, I think that these different functions are convolved to a certain extent.

15/364: "syntactic closures" This is what is being studied here. The closure of (larger) syntactic units. Not hierarchy, not dependency. This is a good term that should be used throughout the manuscript.

19/443f: In line with my above comments, I strongly recommend not using *dependency* closure as target for defining the termination of (large) syntactic units (since this definition will result in many events that are not in fact phrase / constituent closures). The authors might want to have a look at Abney's / Anderson's work to consider the definition of the termination of large syntactic units based on dependency annotations:

Abney, Steven P. „Parsing By Chunks“. In Principle-based parsing, 257–78. Springer, 1991. https://doi.org/10.1007/978-94-011-3474-3_10.

Anderson, Mark, David Vilares, Carlos Gómez-Rodríguez, und Carlos Gómez-Rodríguez. „Artificially Evolved Chunks for Morphosyntactic Analysis“. In Proceedings of the 18th International Workshop on Treebanks and Linguistic Theories (TLT, SyntaxFest 2019), 133–43. Paris, France: Association for Computational Linguistics, 2019. <https://doi.org/10/gm5drk>.

Again, this is to substantiate my earlier comments: I think that the definition of *closure* based on *any* closing dependency does not give reference to the function of prosodic boundaries and the way they relate to syntax: Prosodic boundaries delineate large units (clauses, sentences, heavy NPs, whole VPs), which is also what seems to drive prosodic bootstrapping. Not any dependency closure closes a large syntactic unit that could probably be mapped on a prosodic unit (i.e., every prosodic unit is a syntactic unit, but not any syntactic unit is a prosodic unit).

(3) What it means to be significantly above chance: The classification results I was not particularly impressed by. Classifier performance is barely above chance. While the results are

significant, I still wonder about the meaning of the result. The authors might ask themselves how important / effective a neural / cognitive mechanism is (for language processing and evolution, as stressed by the authors) that only seems to lead to a rather small improvement over and above chance.

10/257: Do I understand correctly that classification accuracy for all conditions was barely above chance (= 50 %)? For "Incoherent", accuracy was "0.507"?

MINOR

5/129: "different sensory modalities" I do not follow. All that has been said so far has mentioned prosody (= acoustic) and abstract (= in the mind / internal rule system / linguistic knowledge). How do "different sensory modalities" come into play? I think there are good reasons for using naturalistic designs (statistical, ecological variance in the signal). Yet, the authors' work does not address multiple sensory modalities I believe. Please rephrase motivation.

6/144f: The authors do not use a model of "cognitive [...] mechanisms". This would be a processing model (e.g., a parser, a working memory model such as cue-based retrieval, etc.). Instead, the authors seem to be using a linguistic theory, assuming that it describes cognitive operations. Please refer to Bever & Poeppel (2010) to understand better the difference between linguistic theory and (computational) models of cognition, as well as their link.

6/151: "representation" or processing? These are different things described by different fields. Linguistic theory is more akin to describing the assumed "representation", whereas the way that the brain arrives at this representation is described by processing models in cognitive science / psycholinguistics. The authors seem to assume that linguistic theory and real-time processing / parsing / cognition and the same thing / exchangeable. Again, please consider Bever & Poeppel (2010) and possibly also Steedman's work.

15/345: "Such a finding is not achievable" This is not correct. An interaction effect in some type of neural response based on a 2 x 2 design (syntactic boundary yes/no, prosodic boundary yes/no) would certainly allow for the same interpretation taken here. Please comment / tone down.

15/346: "information theory" Please explain the link more explicitly. I do not follow from the current sentence. The notion that language has evolved to make communication more efficient

is classic functionalist one that has been around long before information theory was applied / referred to in (psycho)linguistics.

15/359: "predictive mechanisms" Why? prosodic boundaries are audible before the assumed syntactic boundary / end of the syntactic closure (e.g., lengthening, pitch rise). Can the authors be sure that their early effects cannot be explained by such early acoustic markings?

19/443f: I am wondering a bit about the quality of dependency parsing when it comes to audio from "TED talks". Do the authors have an estimate of the quality of the parses?

Reviewer #4 (Remarks to the Author):

Summary & Evaluation

The authors investigate how prosodic phrase boundaries relate to left dependencies (the end of a syntactic phrase) and whether this relation is represented in the brain. They do so by analyzing the speech of four TED talks and reanalyzing an existing MEG dataset with participants who listened to these TED talks. They find that strong prosodic phrase boundaries are more likely to be associated with left-dependencies than weak prosodic phrase boundary. Moreover, the authors trained a decoder on the MEG dataset to separate words with and without left dependencies, showing that the decoder's accuracy improved when words with left dependencies were accompanied by a strong prosodic boundary. While similar findings have been obtained before, this project's contribution extends these findings to a naturalistic dataset. This is good work, making incremental progress in our understanding of the interaction between prosody and syntax in speech processing. There are several assumptions and statistical quirks (e.g., p-values below .05 combined with confidence intervals that include 0) that should be addressed before publishing, so I recommend revise and resubmit.

Major points:

- You train the MEG decoder on weak prosodic boundary data (the neutral group), and then apply it to the neutral, incoherent, and coherent groups, showing that accuracy is highest for the coherent group. However, the linking function here is not clear: if the decoder is trained on the neutral group, why does it perform better on another group? It would only make sense if you assume that strong prosodic boundaries keep the same representation of left dependencies as for the weak prosodic boundaries but make it more extreme, easier to classify. I don't know whether this assumption is correct or not, but you should discuss what it means for the decoder to perform better on the coherent group than on the group it was trained on. You call this neural boosting, but that too is vague: Facilitated processing of left dependencies? Sharper contrasts

in MEG signal between words with and without left dependencies? It would be helpful if you address this explicitly in the manuscript.

- I tried but couldn't understand the analysis with the simulation of the different chi-squared tests: it seems that you are comparing p-values of chi-squared tests on simulated data that assume a 5% frequency shift to the p-values of the chi-squared tests on your data and infer that stronger prosodic boundaries for word with left dependencies than for words without left dependencies (making two separate comparisons based on p-value magnitudes). What you are doing seems cumbersome and unclear to me (and potentially not entirely statistically warranted) given two alternative simple analyses that come to mind that are more traditional and make the point you want to make. I suggest you conduct either one or both instead of the current analysis you have:

o As I see it, you are simply interested in whether there is a difference in prosodic boundary strength between words with and without left dependencies. This sounds like a simple logistic regression: the two binary variables (prosodic boundary strength, left dependencies) can be dummy coded and then you would predict one from the other (e.g., `glm(prosodic boundary ~ left dependency, family=binomial())`).

o More generally, this analysis answers a slightly different question than what you write: it answers whether the presence of left dependencies corresponds to how you binned prosodic boundary strength. Since prosodic boundary strength is continuous, the way to investigate the question of whether left dependencies are associated with an increase in prosodic boundary strength is to keep the measure of strength continuous (before applying the mixture Gamma model) and fitting again a simple Gamma regression (e.g., `glm(prosodic boundary ~ left dependency, family=Gamma())`).

- MVPA decoding: for neutral and incoherent conditions, you report p-values less than .05 but CIs include 0 (in both pre- and post-offset). However, if the CI is a 95% confidence interval, then this is impossible. P-values under .05 always have a 95% CI that does not include 0. If the p-values and confidence intervals were generated in different ways, please explain how and why you rely on the p-value, not the CI, to indicate whether the results are statistically significant.

- For the post-offset time window, the incoherent condition has a p-value greater than .05 and the CI includes 0, yet you claim that the model performed better than chance. This is false: the model did not perform statistically-significantly better than chance.

Minor Points

- For Figure 1, right panel, it would be useful to see not only the proportions of each word group, but also raw counts (either as a second y-axis, as text on the bars, or in whatever way you choose).

- While you talk about prosodic boundary strength, it sounds like you are computing prosodic strength for every word, not necessarily prosodic phrase boundaries. Therefore, it might be clearer if you call this variable prosodic strength/prominence, rather than prosodic boundary strength. If you choose to stay with your terminology as is, I recommend clarifying this point in the beginning (e.g., not all words are actually phrase boundaries, but we computed the prosodic strength and we expect it to increase at phrase boundaries).

- You call your machine learning techniques “state-of-the-art” but I think that is not a very useful term: it does not contribute information about the method, it is debatable what “state-of-the-art” is, and the field is changing so fast that nothing stays “state-of-the-art” for a long time.
- Coherence: “The gamma-mixture model was fitted to the prosodic boundary strength distributions across all 4 TED talks.” This is the first time you mention the number of talks that you had in the dataset, so it was confusing to see it as background information in this sentence. It would be useful to mention the number of talks before this sentence.
- The sentence “The final training and testing sets were composed of 3740 words and 654 words respectively” was confusing: is it correct that there was one training set, and three testing sets? And each testing set had 654 words? It would be good to introduce some redundancy into the sentence (e.g., “the training set was composed of 3740 words and the three testing sets were composed of 654 words each.”).
- I couldn’t understand what analysis was conducted and what the max/inv/train mean in the following sentence: “The coherent decoding was also found to be better in both neutral and incoherent conditions (Max>Inv: pVal = 0.002; Max>Train: pVal < 0.001).”
- Citation 57 (the creators of the dataset) should be credited more prominently (abstract, and when it is mentioned for the first time)
- This sentence is hard to read because of the two consecutive (i.e.,) parentheticals, one of which is very long: “Interestingly, the brain decoding of the words that contained mismatching syntax-prosody combinations (i.e. left dependency but weak prosodic boundary strength, or no left dependency and strong prosodic boundary strength) (i.e. incoherent conditions) showed above-chance performance until word offset, but not afterward”

Typos

- Missing percent sign at the end of “the percentage of words without a left dependency (54.63%) was greater than the one with a left dependency (45.37)”
- Inchoerent->incoherent

We'd like to thank the reviewers for their very helpful and constructive comments and suggestions. We have revised the manuscript to address the points that have been raised. In particular, a re-analysis of the syntactic decoding has been done to follow a suggestion of Reviewer #3. While the approach has been kept identical, the syntactic measure has been changed in order to better model phrase boundaries (see new methods and results sections in the manuscript). Thus, in this response letter, we will here forth refer to the decoding of phrase boundaries and not to left dependencies.

Reviewer #1 (Remarks to the Author):

1) The paper "Speech prosody enhances the neural processing of syntax" by Degano et al. is a paper on the issue how different aspects of speech processing are merged, here merely prosody and left dependency using natural speech as provided by TED talks. The paper reads very well, however, I am not fully satisfied with the grouping of conditions into neutral, coherent and incoherent. Furthermore, I am missing the localization within the brain. Where in the brain are the different aspects of speech processed?

We'd first like to explain why we grouped the conditions into neutral, coherent and incoherent, because we can understand that the choice of these groupings may initially appear puzzling (as you also mention in your next point). In order to statistically compare the conditions (i.e. see if we can successfully decode – i.e. predict from the brain data – the presence or absence of the syntactic feature of interest [i.e. phrase boundaries, in the revised analyses, and left dependencies in the original ones]), we needed to group them based on the two levels of syntactic information. This is because we chose the area under the curve (AUC) as a suitable non-parametric criterion-free measure of generalisation, which by its formulation forces two classes as in training (with and without phrase boundaries) and has been used extensively in MEG decoding ^{1,2}. We thus trained the decoder was trained on a neutral trials, in which there is low prosodic boundary strength (PBS), and presence or absence of the syntactic feature, and likewise the test sets had to then be combinations of the 2 by 2 conditions, each including both levels of the syntactic information. The decision to use the AUC measure for our statistics in a sense forced us to intelligently select pairs of conditions, which when compared to each other, jointly allow to show whether or not the neural decoding of syntactic information is better in the presence of higher PBS.

However, as can be seen in Figure 3 of the Supplementary Information (SI), we followed up on this pairwise analysis with analyses on the 4 different generalisation sets separately, and the results here do converge with the interpretations that have been drawn based on the results reported in the paper.

Regarding neural localisation of the effects, as described above, the design of the decoding analysis that we performed was based on the binarization of the prosodic information within the classes of words with and without phrase boundaries. Importantly the linear decoder that was learnt during the training phase was tested in the three generalisation classes (in 'neutral', 'coherent', 'incoherent'). This choice limits the generalisation of the topology of the MEG channels since we can only identify the underlying network that is enhanced if stronger prosodic information is included. In other words, the information regarding the localization of the model's weights is identical across all the three classes. This means that the localization of

the effect would not be informative due to the fact that (1) this is a multivariate approach that exploits patterns of activations not activity per se and (2) the multivariate patterns don't reflect prosodic information but only phrase boundary decoding. We are currently working on extending this study to a new, fMRI dataset, which will allow precisely to localise the effects of interest.

2) Strong prosodic boundaries were observed twice as often for left dependency words as for no left dependency words (70% vs. 30%). For weak prosodic boundaries there was almost no difference.

While the grouping in 'neutral', 'coherent', 'incoherent' certainly is of a general interest, I was asking myself why a classification according to boundary strength and left dependency was not investigated? In the strong boundary condition - which difference can be observed for LD vs. no LD? For LD - what is the effect of the boundary? In case of no LD is the boundary effect the same?

Please see our response to this question above.

As also noted above, we've addressed single class classification in the SI using the normal accuracy metric; the results show that, as predicted, there is better neural encoding of syntax in the presence of higher PBS.

3) Please check the values - they are identical for weak and high prosodic boundary strength.

"We found that while in the weak prosodic boundary strength cluster, the difference in the proportion of left dependencies was below a 5% frequency shift (p-value of our samples = $7.914e-10 > p\text{-value of test distribution} = 6.233 e-14$), the cluster with high prosodic boundary strength showed a strong bias towards words with left dependencies (p-value of our samples = $7.914e-10 < p\text{-value of test distribution} = 6.233 e-14$)."

We thank the reviewer for spotting this typo. Although the statistical results have not changed, the section and the analysis referred to by this comment has been changed extensively.

4) For the MVPA decoding - how was the chance level estimated? The theoretical level of 0.5 seems to be not appropriate as the classification sometimes is below 0.5? See Allefeld et al 2016 and Valente et al 2021 on how to deal with measures like accuracy values statistically. Please revise the statistics - and test the conditions before combining them.

We thank the reviewer for this insight. The AUC has been used extensively in MEG decoding studies as a more reliable measure than classical accuracies for decoding³. We agree with the reviewer that classification can sometimes be below 0.5, so we tested the individual variability of decoding across conditions and time windows with a permutation test (supplementary material). The results suggest that the null of AUC equal to 0.5 seems appropriate.

As also suggested, we additionally performed prevalence tests⁴⁻⁶ to show how typical the decoding accuracies are within participants in the three conditions, please find the results in the SI.

Using the minimum norm estimates to compute the brain activity is fine, of course. However, it would be interesting to play around with the different regions separately. In that case, a beamformer solution could be superior as such a solution can be computed for each region

separately. The solution is independent of the other regions. One could conduct the MVPA analysis for a set of brain regions known to be involved in processing of prosody and syntax. Would that lead to superior classification results?

The decision to opt for a whole-brain analysis was based on the fact that the information decodable from an ROI-based analysis should, in principle, also be decodable from the whole brain data. While it's true that source reconstruction can potentially suppress unrelated noise, it is contingent upon a precise knowledge of the relevant signal's origin. In the context of our study, the ambiguity in identifying the exact sources of relevant syntactic information led us to prioritize a whole brain approach. By analyzing the entire brain, we aim to capture all potential sources of information, avoiding the risk of excluding potentially very important multi-region information.

We acknowledge that this approach comes with certain challenges, but we believe it aligns with the exploratory nature of our investigation into syntactic processing. Moreover, as previously mentioned, the localization and identification of the network underlying the prosody-syntax interaction is one of the primary focuses of our new, ongoing fMRI study.

minor remarks:

Please use abbreviations for the units in correspondance to the SI, e.g. s instead of sec.

pVal = is usually written as p =

Please present results with the precision motivated by the SNR or some error discussion, e.g. a p-value of 7.914e-10 larger than 6.233e-14 could be more effectively reported as 8e-10 and 6e-14

or the percentage of words without left dependency 55% versus with left dependency 45% ... 31% for words without LD and 69% with LD .. Furthermore, the parameters of the Gamma Distribution - are the five digits really relevant?

Inchoerent -> incoherent

Thank you, we've implemented your suggestions/corrections.

Reviewer #2 (Remarks to the Author):

The study presents a very interesting relationship between prosodic characteristics of spoken language and MEG measures of brain activation while people listen to the same language. The methodology seems appropriate, but I do not have the expertise to judge it. The study used data-driven, fully automated classification methods, which I do not know enough about to critique.

First, the results showed a systematic difference in the spectrogram of the speaking data, that shows, using this methodology, a relationship between speakers' production of strong prosodic boundaries and syntactic left dependencies. We know this, but these results, from a novel methodology, converge with previous findings. The main results of this study are the relationships between prosodic patterns in speech and MEG patterns in the brains of listeners to that speech. All the combinations of syntax and prosodic patterns, even the ones labeled as incoherent, were significantly related to MEG patterns, however, consistent with the main hypothesis, the relationship was stronger between the brain patterns and the coherent combinations than with the neutral and incoherent. This was true also for the

measures of temporal decoding.

So basically, this study has shown, with a different methodology, that prosodic patterns are related to syntactic structure and facilitate brain responses. These results are not novel, but present important converging evidence for the importance of prosody, and acoustic characteristic of speech, for syntactic processing. Given the importance of converging data for psycholinguistics, I believe this paper is worthy of publication, with the caveat that I cannot judge this methodology.

Reviewer #3 (Remarks to the Author):

SUMMARY

This is a fine $n = 11$ MEG study on the interaction of prosody and syntax in the human brain. The authors report that brain activity during the ending of a syntactic dependency can be decoded better when this ending coincides with a prosodic boundary.

The findings have potential for publication. I have some comments on the conceptualization, framing, and motivation that might further improve the value of the current work.

MAJOR

(1) Assumptions versus facts in the motivation of the current study: The authors claim that the hierarchy of syntax is a fact, which is not the case. Rather, hierarchy is assumed by certain theories in (psycho)linguistics, but not others. This should be acknowledged and the text, in part, should be rephrased / toned down. Some example comments on this:

Thank you for having raised these points along with the below concrete examples. We will address the examples one by one below where relevant, and here we provide a more global response to 'major comment' #1.

We agree with the reviewer that hierarchy is a (very wide-spread) theoretical assumption. We have revised the manuscript to clarify that we are studying the impact of representational properties (syntactic and prosodic) on neural decoding.

1/16: "hierarchically" This is not a fact, but a theoretical assumption. Please rephrase ("many linguists assume that" or "one prominent theory of syntax assumes"). I also do not understand how hierarchy is required to make language "efficient and robust". What exactly do the authors mean here by "efficient and robust"?

1/32: See my above comment on assumptions versus facts. It is a frequent habit to call hierarchical grammar a fact about language. This is harmful to the neurobiology of language. There are, for instance, many frameworks in (computational) psycholinguistics that do neither assume hierarchy nor explicitly represent it (e.g., cue-based retrieval, dependency locality theory). The authors should rephrase this sentence and stress that the content is based on their belief / theoretical reasoning, not on facts; alternatively, if they insist that

hierarchy is a fact, they should cite and make explicit a piece of evidence that shows that hierarchy is a fact, in a way understandable / transparent to readers from (neuro)biology.

1/32ff: Can the authors give one example where the formation of hierarchical dependencies is required to understand someone? They write here that "syntactic rules" are "fundamental for successful communication". An example would help non-expert readers to understand this notion here.

6/156: Not all dependency grammars "create hierarchical syntactic representations". Consider, for instance, Nivre's recent work and the experimental work by McElree on the insensitivity of memory retrieval to order. Please rephrase.

We have revised the terminology in the manuscript and importantly changed the syntactic measure (now looking at closed phrase boundaries and not at left dependencies),

15/355: Please explain to the reader how the current findings speak to the processing of "hierarchical structures". Make this explicit, as currently, I think there is no clear link in the text between hierarchies and prosody and boundaries and dependencies.

We no longer use left dependencies in our analyses, so this point is now less relevant. However, in the original version of the manuscript, as is standard terminology both in the context of syntax and computational methods, we used the terms hierarchical structure and dependency structure to refer to syntactic trees. Our calculation of left dependencies was done on a tree and not on the linear word sequence (unlike for example locality dependency theory), and we would therefore still argue that we were operating on hierarchical structures, that is, representations that encode the notion of 'dominance' or 'parenthood' as well as of linear precedence.

(2) Dependency versus boundary: The authors are using the ending of *any* dependency to define the closure of a syntactic unit. I have doubt about this: (1) prosodic boundaries are strong at the end of large syntactic units (e.g., clauses, sentences), but markings at small units are much weaker. (2) Babies bootstrap into syntax from large to small, where only large units can be inferred from acoustics, but small units require statistics. (3) While any prosodic boundary marks a syntactic boundary, not all syntactic boundaries receive a prosodic boundary. (4) Importantly: Not any syntactic dependency coincides with the end of a large syntactic unit. I think that the manuscript in its current form mixes (a) the closure of large syntactic units and (b) the closure of any syntactic dependency. In order to fix this, the authors might think about their definition of syntactic boundary again. I think that there might be a way of defining syntactic boundaries in a more narrow manner that is easier linked to prosody as well, find below some hints on this:

Thank you for having raised these points along with concrete examples. We will address each of the examples below where relevant, and here we provide a more global response to 'major comment' #2

We agree with the reviewer that our results on left dependency might be confounded by a mixture of prosodic markings of syntactic phrase boundaries. We have addressed this issue by changing the syntactic measure of interest in new analyses,

and modelling larger units. To do this we first marked all the tokens in our stimuli that were closing multiple nodes of a parsed tree, as in previous neurolinguistic work with naturalistic stimuli ^{7,8} (see Methods for details).

We then applied the identical methodology that was previously used for left dependencies to this new proxy of phrase boundaries, both for the stimulus analysis and the MEG decoding.

1/23: I am not sure I understand the rationale here. Why would the processing of "prosodic boundaries" affect dependency processing? IPBs mark the on- and offsets of larger syntactic units (clauses, sentences, possibly constituents), but they do not mark dependencies (unless the authors assume that the termination of a dependency is always followed by an IPB, which is certainly not correct in its generality / restricted to head-final constructions and elements occurring at the end of said large syntactic units). Prosodic markings of dependencies are not called prosodic boundaries, these are, for instance, pitch accents that accompany topicalized elements in dependencies. Please make more explicit why and how you think prosodic boundaries relate to dependencies. Edit: From reading the introduction further, I now understand that the authors' work is on the alignment between syntactic and prosodic boundaries (i.e., *phrase*/etc. closure), not dependency processing. Possibly they can rewrite the abstract and first paragraph. Delineating larger syntactic units (such as phrases, clauses, sentences) is very much different from hierarchical syntax (e.g., Christiansen's work, Fodor's work, Frazier's work). There are segments, and then there is fine structure. The prosody–syntax link is likely constrained to the boundaries of large syntactic constituents. The manuscript should be framed in this way I think, rather than jumping the gun with hierarchy (which does not seem to be the objective here).

We thank the reviewer for the comment. We have modified the manuscript accordingly.

2/54: "close correspondence" and "redundancy" This correspondence is not direct and not as strong as the authors want to make the reader believe here (e.g., Truckenbrodt, 1999; Grosejan & Lane, 1979). In particular, prosodic fine structure is debated. It is clear that clauses / sentences / heavy NPs have strong boundaries; beyond these, the evidence is thin. Please be more specific here. There is certainly no full "redundancy".

We thank the reviewer for the comment. We have modified the manuscript to give a more accurate account of the relation.

6/154: "dependency structure" Again, I am not sure how this is supposed to link to prosody. The link between large syntactic units and prosody is where things interact, not in the entire "dependency structure". Phrasing, chunking, clauses, sentences, constituents...How would the authors define the syntactic boundaries of these abstract units based on the "dependency structure"? Maybe the authors could make a figure of an attachment ambiguity to explain to the reader (a) why hierarchy matters, (b) how dependency and prosody come together?

7/177: From the last paragraph of the introduction, it seems to me that the authors assume that "prosodic boundaries" mark all "dependencies" / the termination of all "dependencies". This is quite unusual, in my view. Prosodic boundaries delineate large syntactic units, such

as clauses, sentences, heavy NPs. Prosodic boundaries do not mark any individual dependency (see above comments).

We agree with the reviewer regarding the vague definition. We indeed do not want to suggest that prosodic boundaries mark *all* individual dependencies, our results show in fact that a lower proportion of words with a left dependency are marked. The goal was to show that the occurrence is more likely to appear in the case of a LD, and that the strength of the formed relation is modulated in the brain by prosody.

In the revision, we nonetheless now model closed nodes, which we believe are a better marker for large syntactic units.

7/183: What do the authors mean by "robustness"? What particular aspect of the "representations" is increased in "robustness"? From what has been written, it seems that the authors propose that dependencies are somehow processed more readily when the left-hand element coincides with a prosodic boundary? Again, I am quite confused by what the authors think the function of prosodic boundaries is. To my understanding of the literature, prosodic boundaries mark mostly large syntactic boundaries. So, in my view, the authors should be modeling syntactic boundaries (e.g., of clauses, phrases, large NPs, chunks...) rather than dependencies.

8/198: "This binarization" Again, I am puzzled: Where is the idea coming from that prosodic boundaries mark "left dependencies"? Prosodic boundaries mark *boundaries* of (large) syntactic units. There is literature on prosody marking dependencies (e.g., topicalization, information-structurally driven), but that type of prosody is not prosodic boundaries, but pitch accent / focus etc. The authors' analysis here likely picks up stress / focus rather than prosodic boundaries.

8/218: What is the basis of the idea that "prosodic boundaries" coincide with "left syntactic dependencies"? Prosodic boundaries mark on- and offsets of units, not individual dependencies.

13/294: Framing of study is unclear to me. The authors need to spell out earlier what they think the relationship is between prosodic boundaries and syntactic dependencies. In the current version, I find this link not well motivated. In my view, the broader literature shows that prosodic boundaries mark the boundaries of (larger) syntactic units. Dependencies are not marked by prosodic boundaries. I suspect that what the authors observe in their results is a mixture of prosodic markings of syntactic phrase boundaries (= prosodic boundaries) and prosodic markings of NADs (= pitch accents, intonational focusing). This could be one way of reanalyzing the data: (1) dissociating prosodic boundaries and pitch accents, (2) dissociating syntactic boundaries and NAD onsets. In the current state, I think that these different functions are convolved to a certain extent.

We thank the reviewer for these comments. Indeed, left dependencies are often aligned with syntactic phrase boundaries. Although we originally chose to study left dependencies, in the new version of the manuscript we extracted the number of closed nodes as a proxy for defining the phrase boundary. This approach, as predicted by the reviewer, showed a similar pattern in the stimuli as did left dependencies.

15/364: "syntactic closures" This is what is being studied here. The closure of (larger) syntactic units. Not hierarchy, not dependency. This is a good term that should be used throughout the manuscript.

19/443f: In line with my above comments, I strongly recommend not using *dependency* closure as target for defining the termination of (large) syntactic units (since this definition will result in many events that are not in fact phrase / constituent closures). The authors might want to have a look at Abney's / Anderson's work to consider the definition of the termination of large syntactic units based on dependency annotations:

Abney, Steven P. „Parsing By Chunks“. In Principle-based parsing, 257–78. Springer, 1991. https://doi.org/10.1007/978-94-011-3474-3_10.

Anderson, Mark, David Vilares, Carlos Gómez-Rodríguez, und Carlos Gómez-Rodríguez. „Artificially Evolved Chunks for Morphosyntactic Analysis“. In Proceedings of the 18th International Workshop on Treebanks and Linguistic Theories (TLT, SyntaxFest 2019), 133–43. Paris, France: Association for Computational Linguistics, 2019. <https://doi.org/10/gm5drk>.

Again, this is to substantiate my earlier comments: I think that the definition of *closure* based on *any* closing dependency does not give reference to the function of prosodic boundaries and the way they relate to syntax: Prosodic boundaries delineate large units (clauses, sentences, heavy NPs, whole VPs), which is also what seems to drive prosodic bootstrapping. Not any dependency closure closes a large syntactic unit that could probably be mapped on a prosodic unit (i.e., every prosodic unit is a syntactic unit, but not any syntactic unit is a prosodic unit).

(3) What it means to be significantly above chance: The classification results I was not particularly impressed by. Classifier performance is barely above chance. While the results are significant, I still wonder about the meaning of the result. The authors might ask themselves how important / effective a neural / cognitive mechanism is (for language processing and evolution, as stressed by the authors) that only seems to lead to a rather small improvement over and above chance.

While we agree with the reviewer that the AUC values are low, our approach and therefore the statistical analysis are in line with the values of classification accuracies of abstract elements of natural language obtained in previous studies^{9–11}.

To further validate our decoding approach and the replicability of the effects at the individual level, we performed additional prevalence testing analyses, reported in the Supplementary Information. The prevalence scores that were obtained in the decoding confirm the robustness of our results, even with low numerical AUC values .

10/257: Do I understand correctly that classification accuracy for all conditions was barely above chance (= 50 %)? For "Incoherent", accuracy was "0.507"?

This analysis has been changed in the new manuscript

MINOR

We thank the reviewer for these minor comments. All of them have been addressed and changed in the manuscript.

5/129: "different sensory modalities" I do not follow. All that has been said so far has mentioned prosody (= acoustic) and abstract (= in the mind / internal rule system / linguistic knowledge). How do "different sensory modalities" come into play? I think there are good reasons for using naturalistic designs (statistical, ecological variance in the signal). Yet, the authors' work does not address multiple sensory modalities I believe. Please rephrase motivation.

6/144f: The authors do not use a model of "cognitive [...] mechanisms". This would be a processing model (e.g., a parser, a working memory model such as cue-based retrieval, etc.). Instead, the authors seem to be using a linguistic theory, assuming that it describes cognitive operations. Please refer to Bever & Poeppel (2010) to understand better the difference between linguistic theory and (computational) models of cognition, as well as their link.

6/151: "representation" or processing? These are different things described by different fields. Linguistic theory is more akin to describing the assumed "representation", whereas the way that the brain arrives at this representation is described by processing models in cognitive science / psycholinguistics. The authors seem to assume that linguistic theory and real-time processing / parsing / cognition and the same thing / exchangeable. Again, please consider Bever & Poeppel (2010) and possibly also Steedman's work.

We thank the reviewer for highlighting this inconsistency. We have changed the manuscript accordingly.

15/345: "Such a finding is not achievable" This is not correct. An interaction effect in some type of neural response based on a 2 x 2 design (syntactic boundary yes/no, prosodic boundary yes/no) would certainly allow for the same interpretation taken here. Please comment / tone down.

15/346: "information theory" Please explain the link more explicitly. I do not follow from the current sentence. The notion that language has evolved to make communication more efficient is classic functionalist one that has been around long before information theory was applied / referred to in (psycho)linguistics.

15/359: "predictive mechanisms" Why? prosodic boundaries are audible before the assumed syntactic boundary / end of the syntactic closure (e.g., lengthening, pitch rise). Can the authors be sure that their early effects cannot be explained by such early acoustic markings?

19/443f: I am wondering a bit about the quality of dependency parsing when it comes to audio from "TED talks". Do the authors have an estimate of the quality of the parses?

The final parser has been modified (Berkeley Neural Parser in Spacy). Although errors may be present in the automatic parsing of these trees, we believe that annotating such a specific metric of syntax (i.e. closed nodes) will not introduce enough noise into the cortical classifier to alter the overall trends seen in our results. Due to our

randomised cross-validation approach, it can be assumed that these errors should be evenly distributed across all conditions and thus not affect the overall pattern of results, and the related interpretation of these.

Reviewer #4 (Remarks to the Author):

Summary & Evaluation

The authors investigate how prosodic phrase boundaries relate to left dependencies (the end of a syntactic phrase) and whether this relation is represented in the brain. They do so by analyzing the speech of four TED talks and reanalyzing an existing MEG dataset with participants who listened to these TED talks. They find that strong prosodic phrase boundaries are more likely to be associated with left-dependencies than weak prosodic phrase boundary. Moreover, the authors trained a decoder on the MEG dataset to separate words with and without left dependencies, showing that the decoder's accuracy improved when words with left dependencies were accompanied by a strong prosodic boundary. While similar findings have been obtained before, this project's contribution extends these findings to a naturalistic dataset. This is good work, making incremental progress in our understanding of the interaction between prosody and syntax in speech processing. There are several assumptions and statistical quirks (e.g., p-values below .05 combined with confidence intervals that include 0) that should be addressed before publishing, so I recommend revise and resubmit.

Major points:

- You train the MEG decoder on weak prosodic boundary data (the neutral group), and then apply it to the neutral, incoherent, and coherent groups, showing that accuracy is highest for the coherent group. However, the linking function here is not clear: if the decoder is trained on the neutral group, why does it perform better on another group? It would only make sense if you assume that strong prosodic boundaries keep the same representation of left dependencies as for the weak prosodic boundaries but make it more extreme, easier to classify. I don't know whether this assumption is correct or not, but you should discuss what it means for the decoder to perform better on the coherent group than on the group it was trained on. You call this neural boosting, but that too is vague: Facilitated processing of left dependencies? Sharper contrasts in MEG signal between words with and without left dependencies? It would be helpful if you address this explicitly in the manuscript.

We agree with the reviewer that the interpretation of the AUC increase (or decrease) across the three conditions was unclear. Indeed we interpret our results in the coherent condition as an effect of sharpening of the MEG features thus improving the decoding results.

- I tried but couldn't understand the analysis with the simulation of the different chi-squared tests: it seems that you are comparing p-values of chi-squared tests on simulated data that assume a 5% frequency shift to the p-values of the chi-squared tests on your data and infer that stronger prosodic boundaries for word with left dependencies than for words without left dependencies (making two separate comparisons based on p-value magnitudes). What you are doing seems cumbersome and unclear to me (and potentially not entirely statistically warranted) given two alternative simple analyses that come to mind that are more traditional and make the point you want to make. I suggest you conduct either one or both instead of the current analysis you have:

- o As I see it, you are simply interested in whether there is a difference in prosodic boundary strength between words with and without left dependencies. This sounds like a simple logistic regression: the two binary variables (prosodic boundary strength, left dependencies) can be dummy coded and then you would predict one from the other (e.g., `glm(prosodic boundary ~ left dependency, family=binomial())`).
- o More generally, this analysis answers a slightly different question than what you write: it answers whether the presence of left dependencies corresponds to how you binned prosodic boundary strength. Since prosodic boundary strength is continuous, the way to investigate the question of whether left dependencies are associated with an increase in prosodic boundary strength is to keep the measure of strength continuous (before applying the mixture Gamma model) and fitting again a simple Gamma regression (e.g., `glm(prosodic boundary ~ left dependency, family=Gamma())`).

We thank the reviewer for the comment. Considering the high amount of samples/words, our first idea was to assess the imbalance across the two classes by using an arbitrary threshold of 5%. In other words, the simulation allowed us to test for whether the difference in the occurrence of prosodic boundaries in the two left dependency classes was sensible enough. We agree that this reasoning may have been too cumbersome, and in the revision have now therefore run the analysis suggested by the reviewer, and report these in the revised manuscript

- MVPA decoding: for neutral and incoherent conditions, you report p-values less than .05 but CIs include 0 (in both pre- and post-offset). However, if the CI is a 95% confidence interval, then this is impossible. P-values under .05 always have a 95% CI that does not include 0. If the p-values and confidence intervals were generated in different ways, please explain how and why you rely on the p-value, not the CI, to indicate whether the results are statistically significant. is false: the model did not perform statistically-significantly better than chance.

We thank the reviewer for spotting this mistake. There was indeed a problem with the computation of the confidence intervals and we thank the reviewer for noticing. Since the distribution is non-Gaussian, we recomputed the p-values with a non parametric test, and have thus removed the CI in the new analysis. We also corrected the paragraph according to the new analysis.

- Minor Points]

We thank the reviewer for these minor comments. All of them have been addressed and changed in the manuscript.

- For Figure 1, right panel, it would be useful to see not only the proportions of each word group, but also raw counts (either as a second y-axis, as text on the bars, or in whatever way you choose).

- While you talk about prosodic boundary strength, it sounds like you are computing prosodic strength for every word, not necessarily prosodic phrase boundaries. Therefore, it might be clearer if you call this variable prosodic strength/prominence, rather than prosodic boundary strength. If you choose to stay with your terminology as is, I recommend clarifying this point in the beginning (e.g., not all words are actually phrase boundaries, but we computed the prosodic strength and we expect it to increase at phrase boundaries).

We have added this clarification in the introduction and in the methods section.

- You call your machine learning techniques “state-of-the-art” but I think that is not a very useful term: it does not contribute information about the method, it is debatable what “state-of-the-art” is, and the field is changing so fast that nothing stays “state-of-the-art” for a long time.

If this is about parsing one could say the following: the parser we use now has a published accuracy of 95% correct. This is state-of-the-art according to current leaderboards, and if other parsers were to appear with better results, they are likely to be minimal improvements, given that parsing performance has been around 90-95% for the last decade.

- Coherence: “The gamma-mixture model was fitted to the prosodic boundary strength distributions across all 4 TED talks.” This is the first time you mention the number of talks that you had in the dataset, so it was confusing to see it as background information in this sentence. It would be useful to mention the number of talks before this sentence.

- The sentence “The final training and testing sets were composed of 3740 words and 654 words respectively” was confusing: is it correct that there was one training set, and three testing sets? And each testing set had 654 words? It would be good to introduce some redundancy into the sentence (e.g., “the training set was composed of 3740 words and the three testing sets were composed of 654 words each.”).

- I couldn’t understand what analysis was conducted and what the max/inv/train mean in the following sentence: “The coherent decoding was also found to be better in both neutral and incoherent conditions (Max>Inv: pVal = 0.002; Max>Train: pVal < 0.001).”

- Citation 57 (the creators of the dataset) should be credited more prominently (abstract, and when it is mentioned for the first time)

We thank the reviewer for the feedback. Due to journal guidelines we didn’t add any references to the abstract, but we have made sure that the TEDium MEG dataset has been cited when mentioned the first time.

- This sentence is hard to read because of the two consecutive (i.e.,) parentheticals, one of which is very long: “Interestingly, the brain decoding of the words that contained mismatching syntax-prosody combinations (i.e. left dependency but weak prosodic boundary strength, or no left dependency and strong prosodic boundary strength) (i.e. incoherent conditions) showed above-chance performance until word offset, but not afterward”

Typos

- Missing percent sign at the end of “the percentage of words without a left dependency (54.63%) was greater than the one with a left dependency (45.37)”

- Inchoerent->incoherent

References

1. King, J.-R. & Dehaene, S. Characterizing the dynamics of mental representations: the temporal generalization method. *Trends Cogn. Sci.* **18**, 203–210 (2014).
2. Gwilliams, L., King, J.-R., Marantz, A. & Poeppel, D. Neural dynamics of phoneme sequences reveal position-invariant code for content and order. *Nat. Commun.* **13**, 6606 (2022).
3. King, J.-R. & Dehaene, S. Characterizing the dynamics of mental representations: the temporal generalization method. *Trends Cogn. Sci.* **18**, 203–210 (2014).
4. Ince, R. A. A., Paton, A. T., Kay, J. W. & Schyns, P. G. Bayesian inference of population prevalence. (2021) doi:10.7554/eLife.62461.
5. Allefeld, C., Görgen, K. & Haynes, J.-D. Valid population inference for information-based imaging: From the second-level t-test to prevalence inference. *Neuroimage* **141**, 378–392 (2016).
6. Donhauser, P. W., Florin, E. & Baillet, S. Imaging of neural oscillations with embedded inferential and group prevalence statistics. *PLoS Comput. Biol.* **14**, e1005990 (2018).
7. Li, J. & Hale, J. Grammatical predictors for fMRI time-courses. in (Oxford University Press, 2019).
8. Nelson, M. J. *et al.* Neurophysiological dynamics of phrase-structure building during sentence processing. *Proc. Natl. Acad. Sci. U. S. A.* **114**, E3669–E3678 (2017).
9. Gwilliams, L., Marantz, A., Poeppel, D. & King, J.-R. Top-down information shapes lexical processing when listening to continuous speech. *Language, Cognition and Neuroscience* (2023) doi:10.1080/23273798.2023.2171072.
10. Gwilliams, L., King, J.-R., Marantz, A. & Poeppel, D. Neural dynamics of phoneme

sequences reveal position-invariant code for content and order. *Nat. Commun.* **13**, 6606 (2022).

11. Gennari, G., Marti, S., Palu, M., Fló, A. & Dehaene-Lambertz, G. Orthogonal neural codes for speech in the infant brain. *Proc. Natl. Acad. Sci. U. S. A.* **118**, (2021).

Reviewers' comments:

Reviewer #1 (Remarks to the Author):

I would like to thank the authors for carefully answering my comments. I have no further comments to make.

Reviewer #4 (Remarks to the Author):

The revision has substantially improved the original submissions. Reiterating my original review, this is a well-done study, which, while not providing a novel theoretical contribution, offers a novel method of examining the relationship between syntactic boundaries and prosodic phrase boundaries (using decoding of MEG data with naturalistic stimuli). If the manuscript were to be published as-is, it would be all right. I still have some comments, the most major one addressing the linking function between the findings and a theory of sentence processing; I think the paper would become better if the authors addressed those comments.

Major

- It might be useful to address explicitly the linking function of your findings and sentence processing (as mentioned in my original review). Your finding is nuanced: a decoder trained on MEG data with weak prosodic boundaries is more accurate at whether a syntactic boundary was present when the prosodic structure aligns with the syntactic structure than if they're misaligned. Because decoding is a black box, the opposite finding (incoherent > coherent) would lead you to the same conclusion. In other words, in my understanding, all you can say from your findings is that we found that prosodic phrase boundaries "matter" in sentence processing.

o In your discussion you write "We tested the hypothesis that the brain represents closing phrase boundaries differently depending on the prosodic structure of the sentences." However, "represent differently" implies to me that processing happens somehow differently in time or space, that there is a categorically different representation for phrase boundaries. However, if, for instance, the difference is in strength of activation, "represents differently" isn't an accurate description (the representation is the same, the strength of activation is changed). For example, maybe the brain doesn't represent closing phrase boundaries at all, and instead it is the case that what we analyze as a phrase boundary is simply a point where processing is more costly. This is all to say that you would be served by an even more general phrasing, like "We tested the hypothesis that sentence processing at syntactic phrase boundaries is moderated by the prosodic structure of the sentence."

o "suggesting that the presence of prosodic cues leads to enhanced syntactic processing." This is not necessarily correct. All you know is that model that was trained on no-prosody data did better with congruent prosody (and you would arrive at the same conclusions even with an opposite data pattern). For example, it could be that incongruent prosody was adding noise (a-la N400) and congruent prosody didn't enhance processing, but simply removed noise. This goes back to the idea that decoding can only tell that the processing in two conditions is not the same, but nothing more specific.

Minor

- "Syntactic rules define syntactic structure, which can give rise to an infinite number of utterances." You are right to point out that language is a combinatorial system and that there are very many possible utterances. Although the claim about language having infinitely many possible utterances is theoretically important for certain theories of language (e.g., Chomsky, Hauser, and Fitch, 2002), it is inaccurate: the number of words in a language is finite, and words cannot combine infinitely (e.g., beyond the number of the atoms in the universe). The critical point is that compositionality gives us compression: instead of having a symbol for any possible meaning we might want to convey, we have relatively few symbols (something like words) that we can combine in many ways (syntax). The compression is the important part here, not infinity per se.

- The following statement is problematic: "To do this we changed the link function to a gamma function(family = Gamma()) of the dplyr package in R."

o The dplyr package is used for data wrangling, not for modeling, so this function doesn't come from there. The package that has the distribution family is stats and you load it implicitly when you load lme4.

o The linking function is the inverse (that's the default, which you are using based on the specification above); the distribution family is Gamma. So, in other words a more accurate statement would be To do this we used a generalized linear model from the Gamma family using the inverse linking function(cite lme4/stats).

- "Based on these results, we created three different groups of word tokens (with resampling) that could be evaluated in the brain decoding" This phrasing was confusing to me. If my understanding is correct, that by resampling you mean that there is overlap between the groups, maybe it can be phrased accordingly (e.g., when describing the second group, adding a sentence saying that stimuli with weak prosodic boundary and no phrase boundary figured twice, in group 1 and in group 2.). At least to me, "resampling" sounds like you were using some kind of bootstrap.

- "A second group included words without a closing phrase boundary and a weak prosodic boundary, as well as words with a closing phrase boundary and a strong prosodic boundary, called here 'coherent', since this group included the most statistically probable combination based on the stimuli analysis." Coherent is not the same as most frequent; I think you call this coherent because that's what you expect to co-occur theoretically (you predict that a strong prosodic boundary is produced when a closing phrase boundary is present, e.g., to facilitate processing/production). (This is not a critical point, I just thought that the word "coherent" was a great choice and that your explanation for why you use it doesn't hit the nail on the head). The same rationale applies to "incoherent" and group 3.

- "The decoding of the coherent condition was found to be better than that of the neutral and incoherent conditions (Coherent>Incoherent: $p=0.004$; Coherent>Neutral: $p=0.014$)." How did you get those comparisons? Some kind of post-hoc comparison? In which case, how did you correct for multiple comparisons?

- Top-right panel of Figure 2 is very useful, but can be done a bit clearer, in my opinion. Perhaps, for strong/weak, maybe add "Pr" to indicate it's prosody? And the blue/red ellipses are so broad that it might look like they cover all quadrants: maybe a figure 8 would look better?

- Results: Temporal Decoding. I think this section would benefit from adding one more sentence in the beginning that would orient the reader to what this analysis means; your first sentence mentions statistical significance, but it would be good to first set up what we are looking at with this test, before we say it's significant (regardless of what you do in the method section, since many might not read it).

o Going off of that, maybe another beneficial sentence would be telling us what temporal decoding tells us that MVPA decoding didn't (not how they are different methodologically. How do they differ in what they mean?). You have a good one in the discussion: "The second method made use of temporal decoding, to give us a finer-grained and continuous view of the dynamics of potential phrase boundary and prosodic boundary interactions."

- Results: Temporal Decoding. You write "we found that the presence of prosodic content boosted," but that's inaccurate (and I was confused). There is always prosodic content present in speech. The coherent condition is unique in that the prosody and the syntax matched in a theoretically meaningful way (going back to why you are calling this the coherent condition). When prosody and syntax are coherent (I love your choice of this term, it's wonderfully appropriate in this context) we see more successful decoding. (and, like my major comment, we don't know whether boosting is what happened; moreover, what is boosting? More blood flow to neurons? Less blood flow to neurons because processing is facilitated? I'd be careful with the claim you are making.)

- To do this we ran a non-parametric cluster-level paired t-test with 2046 permutations () ->

$2^{11} = 2048$. Are you subtracting two because you are adjusting the degrees of freedom? I was confused to see $2^{11} = 2046$.

Typos:

- No space in Figure 2 caption: "The left panel corresponds to the decoding in the pre-stimulus offset time window In both pre- and post-offset data"
- Maybe missing a determiner in the Figure 3 caption "the decoding of closing phrase boundaries in coherent condition" -> in THE coherent condition
- Figure 3 caption: "Coherent and neutral sets performed performed above chance" -> The word "performed" repeats twice
- Figure 3 caption: "strength. \ Horizontal red lines represent significant results under cluster-based permutation tests." -> misplaced \ sign?

Reviewer #5 (Remarks to the Author):

I thank the authors for addressing the whole comments about dependency (i.e., define the syntactic boundary from the closing nodes now). I think the conceptualization/framing makes sense now. Below are some minor suggestions.

(1) Maybe refer to the result figures in the method section. (e.g. refer to the Fig.1 in the section of Assessing the prosody-syntax interaction in the stimuli).

(2) Fig. 4: since the authors had the part-of-speech and then parsed the sentence, it would be clear if the authors also add part-of-speech labelling for each word in the parsed tree here.

(3) In MVPA statistics (Page 21), maybe explicitly mention what you replied in the letter: "To further validate our decoding approach and the replicability of the effects at the individual level, we performed additional prevalence testing analyses, reported in the Supplementary Information. The prevalence scores that were obtained in the decoding confirm the robustness of our results, even with low numerical AUC values."

(4) The authors might need to check typos/additional space/repetitive words in the whole manuscript.

typo:

Page 2: "Jonny, the little boy is playing" → Johnny

Page 7: a closing phrase boundary y and → an extra y?

Fig. 1: left plot: x label: prosodic boundary strengt (DN) → strength
by the way: what does "DN" mean here?

Fig. 4: using gamma mixtures models → using gamma mixture models?

Second round

Reviewer #4 (Remarks to the Author):

The revision has substantially improved the original submissions. Reiterating my original review, this is a well-done study, which, while not providing a novel theoretical contribution, offers a novel method of examining the relationship between syntactic boundaries and prosodic phrase boundaries (using decoding of MEG data with naturalistic stimuli). If the manuscript were to be published as-is, it would be all right. I still have some comments, the most major one addressing the linking function between the findings and a theory of sentence processing; I think the paper would become better if the authors addressed those comments.

Major

- It might be useful to address explicitly the linking function of your findings and sentence processing (as mentioned in my original review). Your finding is nuanced: a decoder trained on MEG data with weak prosodic boundaries is more accurate at whether a syntactic boundary was present when the prosodic structure aligns with the syntactic structure than if they're misaligned. Because decoding is a black box, the opposite finding (incoherent > coherent) would lead you to the same conclusion. In other words, in my understanding, all you can say from your findings is that we found that prosodic phrase boundaries "matter" in sentence processing.

o In your discussion you write "We tested the hypothesis that the brain represents closing phrase boundaries differently depending on the prosodic structure of the sentences." However, "represent differently" implies to me that processing happens somehow differently in time or space, that there is a categorically different representation for phrase boundaries. However, if, for instance, the difference is in strength of activation, "represents differently" isn't an accurate description (the representation is the same, the strength of activation is changed). For example, maybe the brain doesn't represent closing phrase boundaries at all, and instead it is the case that what we analyze as a phrase boundary is simply a point where processing is more costly. This is all to say that you would be served by an even more general phrasing, like "We tested the hypothesis that sentence processing at syntactic phrase boundaries is moderated by the prosodic structure of the sentence."

We thank the reviewer for this feedback, we agree and have adjusted accordingly.

o "suggesting that the presence of prosodic cues leads to enhanced syntactic processing." This is not necessarily correct. All you know is that model that was trained on no-prosody data did better with congruent prosody (and you would arrive at the same conclusions even with an opposite data pattern). For example, it could be that incongruent prosody was adding noise (a-la N400) and congruent prosody didn't enhance processing, but simply removed

noise. This goes back to the idea that decoding can only tell that the processing in two conditions is not the same, but nothing more specific.

We agree with the reviewer that incongruent prosody can potentially add enough noise to justify the difference with congruent prosody. However, we believe that the fact that the congruent condition shows greater decodability with the same weights trained with the neutral condition is an indication that this pattern is more consistent with a 'pure' syntactic representation (i.e. containing weak prosodic boundaries). For this reason, we think that the word *enhance* is appropriate even if congruent prosody is 'just' removing noise. That said, we added additional information regarding this point in the discussion as well as clarifying the choice of wording.

Minor

- "Syntactic rules define syntactic structure, which can give rise to an infinite number of utterances." You are right to point out that language is a combinatorial system and that there are very many possible utterances. Although the claim about language having infinitely many possible utterances is theoretically important for certain theories of language (e.g., Chomsky, Hauser, and Fitch, 2002), it is inaccurate: the number of words in a language is finite, and words cannot combine infinitely (e.g., beyond the number of the atoms in the universe). The critical point is that compositionality gives us compression: instead of having a symbol for any possible meaning we might want to convey, we have relatively few symbols (something like words) that we can combine in many ways (syntax). The compression is the important part here, not infinity per se.

**Thanks for this comment, we have reworded this passage as follows:
'Syntactic rules define syntactic structure, which allows to combine the limited number of words contained in the lexicon into a quasi-infinite number of utterances. This compositional property of syntax has an important role in speech comprehension, since it allows to flexibly combine words into coherent syntactic and semantic units, supporting efficient communication'.**

- The following statement is problematic: "To do this we changed the link function to a gamma function(family = Gamma()) of the dplyr package in R)."

o The dplyr package is used for data wrangling, not for modeling, so this function doesn't come from there. The package that has the distribution family is stats and you load it implicitly when you load lme4.

o The linking function is the inverse (that's the default, which you are using based on the specification above); the distribution family is Gamma. So, in other words a more accurate statement would be To do this we used a generalized linear model from the Gamma family using the inverse linking function(cite lme4/stats).

We thank the reviewer for spotting this error and other issues, we have adjusted the respective passages.

- “Based on these results, we created three different groups of word tokens (with resampling) that could be evaluated in the brain decoding” This phrasing was confusing to me. If my understanding is correct, that by resampling you mean that there is overlap between the groups, maybe it can be phrased accordingly (e.g., when describing the second group, adding a sentence saying that stimuli with weak prosodic boundary and no phrase boundary figured twice, in group 1 and in group 2.). At least to me, “resampling” sounds like you were using some kind of bootstrap.

We have added a sentence after the description of the groups to clarify that some tokens were indeed used multiple times.

- “A second group included words without a closing phrase boundary and a weak prosodic boundary, as well as words with a closing phrase boundary and a strong prosodic boundary, called here ‘coherent’, since this group included the most statistically probable combination based on the stimuli analysis.” Coherent is not the same as most frequent; I think you call this coherent because that’s what you expect to co-occur theoretically (you predict that a strong prosodic boundary is produced when a closing phrase boundary is present, e.g., to facilitate processing/production). (This is not a critical point, I just thought that the word “coherent” was a great choice and that your explanation for why you use it doesn’t hit the nail on the head). The same rationale applies to “incoherent” and group 3.

We originally used the term ‘coherent’ to refer to the coherence with the pattern observed in the stimuli, and not necessarily theoretically. But since what we observed in the stimuli is indeed consistent with what we had predicted, we’ve now added ‘and theoretically’ to our statement regarding why we used this term.

- “The decoding of the coherent condition was found to be better than that of the neutral and incoherent conditions (Coherent>Incoherent: $p=0.004$; Coherent>Neutral: $p=0.014$).” How did you get those comparisons? Some kind of post-hoc comparison? In which case, how did you correct for multiple comparisons?

We tested them with paired permutation testing corrected via bonferroni-holm. We added a sentence in the methods section.

- Top-right panel of Figure 2 is very useful, but can be done a bit clearer, in my opinion. Perhaps, for strong/weak, maybe add “Pr” to indicate it’s prosody? And the blue/red ellipses are so broad that it might look like they cover all quadrants: maybe a figure 8 would look better?

We have added additional information to make it clearer. We have also tried to change the ellipses to figure 8, but we are not very satisfied with the result (see figure below). We have thus kept the previous version, but if the reviewer feels that this is significantly better, we will consider this change.

- Results: Temporal Decoding. I think this section would benefit from adding one more sentence in the beginning that would orient the reader to what this analysis means; your first sentence mentions statistical significance, but it would be good to first set up what we are looking at with this test, before we say it's significant (regardless of what you do in the method section, since many might not read it).

o Going off of that, maybe another beneficial sentence would be telling us what temporal decoding tells us that MVPA decoding didn't (not how they are different methodologically. How do they differ in what they mean?). You have a good one in the discussion: "The second method made use of temporal decoding, to give us a finer-grained and continuous view of the dynamics of potential phrase boundary and prosodic boundary interactions."

We added the following: To better understand the temporal dynamics of the representation of phrase boundaries, we also employed temporal decoding across the brain activity before and after word offset (Fig. 3). Indeed, this approach offered a more continuous perspective on the temporal evolution of the interactions between the closing of phrase boundaries and prosodic boundary strength.

- Results: Temporal Decoding. You write "we found that the presence of prosodic content boosted," but that's inaccurate (and I was confused). There is always prosodic content present in speech. The coherent condition is unique in that the prosody and the syntax matched in a theoretically meaningful way (going back to why you are calling this the coherent condition). When prosody and syntax are coherent (I love your choice of this term, it's wonderfully appropriate in this context) we see more successful decoding. (and, like my major comment, we don't know whether boosting is what happened; moreover, what is

boosting? More blood flow to neurons? Less blood flow to neurons because processing is facilitated? I'd be careful with the claim you are making.)

We've reworded this section as follows: 'we found better decoding of syntax in the coherent condition, suggesting that higher prosodic boundary strength is associated with boosting of the strength of the syntactic classification'. Please note that here we wrote that it's the strength of the syntactic classification that's boosted, and not the neural processing. In other places where we do suggest that our findings support the idea that syntactic processing is boosted by prosody, we now qualify this term by adding the word 'possible'. We agree that our results could suggest this but that we have to be cautious in making stronger statements about this.

- To do this we ran a non-parametric cluster-level paired t-test with 2046 permutations () -> $2^{11} = 2048$. Are you subtracting two because you are adjusting the degrees of freedom? I was confused to see $2^{11}=2046$.

We thank the reviewer for spotting this typo; we ran 2048 permutations and have corrected this in the paper.

Typos:

- No space in Figure 2 caption: "The left panel corresponds to the decoding in the pre-stimulus offset time windowIn both pre- and post-offset data"
- Maybe missing a determiner in the Figure 3 caption "the decoding of closing phrase boundaries in coherent condition" -> in THE coherent condition
- Figure 3 caption: "Coherent and neutral sets performed performed above chance" -> The word "performed" repeats twice
- Figure 3 caption: "strength.\ Horizontal red lines represent significant results under cluster-based permutation tests." -> misplaced \ sign?

Thank you for your careful reading; we've corrected these issues too.

Reviewer #5 (Remarks to the Author):

I thank the authors for addressing the whole comments about dependency (i.e., define the syntactic boundary from the closing nodes now). I think the conceptualization/framing makes sense now. Below are some minor suggestions.

Thank you for your positive assessment of our revision.

(1) Maybe refer to the result figures in the method section. (e.g. refer to the Fig.1 in the section of Assessing the prosody-syntax interaction in the stimuli).

Thanks, done.

(2) Fig. 4: since the authors had the part-of-speech and then parsed the sentence, it would be clear if the authors also add part-of-speech labelling for each word in the parsed tree here.

POS labels have been added

(3) In MVPA statistics (Page 21), maybe explicitly mention what you replied in the letter: “To further validate our decoding approach and the replicability of the effects at the individual level, we performed additional prevalence testing analyses, reported in the Supplementary Information. The prevalence scores that were obtained in the decoding confirm the robustness of our results, even with low numerical AUC values.”

Thank you for this suggestion, we’ve added this to the paper.

(4) The authors might need to check typos/additional space/repetitive words in the whole manuscript.

typo:

Page 2: “Jonny, the little boy is playing” → Johnny

Page 7: a closing phrase boundary y and → an extra y?

Fig. 1: left plot: x label: prosodic boundary strengt (DN) → strength
by the way: what does “DN” mean here?

Fig. 4: using gamma mixtures models → using gamma mixture models?

Thank you for your careful reading, we’ve corrected all of these typos/mistakes.

REVIEWERS' COMMENTS:

Reviewer #4 (Remarks to the Author):

The authors have satisfactorily addressed all of my comments (and I think they should keep the original figure that is still currently in the manuscript), but I want to ask them for one more change (I don't need to review it). If possible, please cite R and the packages that you used properly (using citations and bibliography, like one would cite a paper), since this is the most appropriate way to acknowledge their important contribution. You can get those by entering the command "citation()" into the terminal for the citation for R, or "citation(packagename)" for the citation for the package of interest. Below are the citations for R and lme4 in bibtex format that I acquired in this way for you.

R:

```
@Manual{,  
title = {R: A Language and Environment for Statistical Computing},  
author = {{R Core Team}},  
organization = {R Foundation for Statistical Computing},  
address = {Vienna, Austria},  
year = {2023},  
url = {https://www.R-project.org/},  
}
```

lme4:

```
@Article{,  
title = {Fitting Linear Mixed-Effects Models Using {lme4}},  
author = {Douglas Bates and Martin M{"a"}chler and Ben Bolker and Steve Walker},  
journal = {Journal of Statistical Software},  
year = {2015},  
volume = {67},  
number = {1},  
pages = {1--48},  
doi = {10.18637/jss.v067.i01},  
}
```

REVIEWERS' COMMENT

Reviewer #4 (Remarks to the Author):

The authors have satisfactorily addressed all of my comments (and I think they should keep the original figure that is still currently in the manuscript), but I want to ask them for one more change (I don't need to review it). If possible, please cite R and the packages that you used properly (using citations and bibliography, like one would cite a paper), since this is the most appropriate way to acknowledge their important contribution. You can get those by entering the command "citation()" into the terminal for the citation for R, or "citation(packagename)" for the citation for the package of interest. Below are the citations for R and lme4 in bibtex format that I acquired in this way for you.

R:

```
@Manual{  
  title = {R: A Language and Environment for Statistical Computing},  
  author = {{R Core Team}},  
  organization = {R Foundation for Statistical Computing},  
  address = {Vienna, Austria},  
  year = {2023},  
  url = {https://www.R-project.org/},  
}
```

lme4:

```
@Article{  
  title = {Fitting Linear Mixed-Effects Models Using {lme4}},  
  author = {Douglas Bates and Martin M{"a"}chler and Ben Bolker and Steve Walker},  
  journal = {Journal of Statistical Software},  
  year = {2015},  
  volume = {67},  
  number = {1},  
  pages = {1--48},  
  doi = {10.18637/jss.v067.i01},  
}
```

We have modified and added the citation as requested.